# LINFUSION: 1 GPU, 1 MINUTE, 16K IMAGE

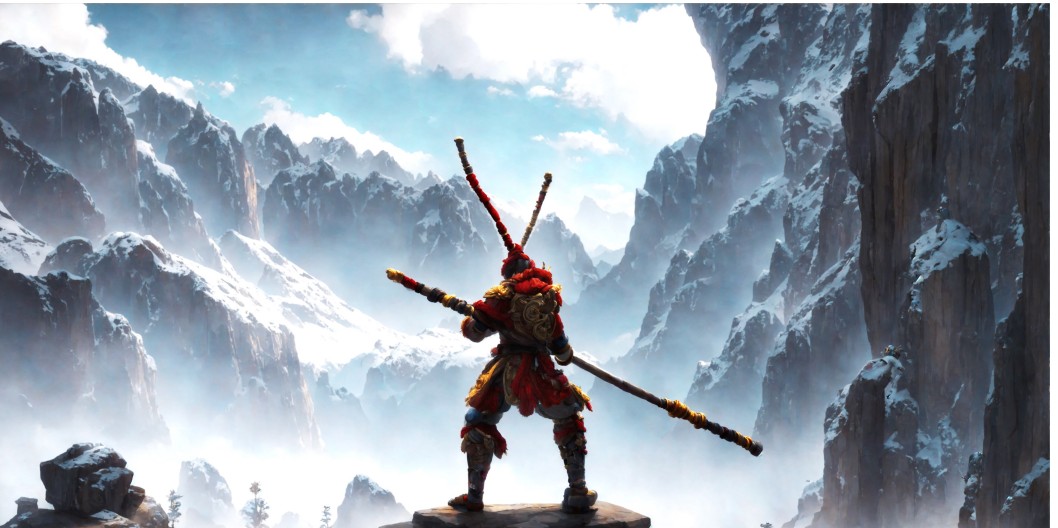

Figure 1: A $16384 \times 8192$-resolution example in the theme of *Black Myth: Wukong* generated by LinFusion on a single GPU with Canny-conditioned ControlNet. The textual prompt is "`the back view of the Monkey King holding a rod in hand stands, 16k, high quality, best quality, style of a 3A game, fantastic style`". The original picture and the extracted Canny edge are shown in Fig. 5.

## ABSTRACT

Modern diffusion models, particularly those utilizing a Transformer-based UNet for denoising, rely heavily on self-attention operations to manage complex spatial relationships, thus achieving impressive generation performance. However, this existing paradigm faces significant challenges in generating high-resolution visual content due to its quadratic time and memory complexity with respect to the number of spatial tokens. To address this limitation, we aim at a novel linear attention mechanism as an alternative in this paper. Specifically, we begin our exploration from recently introduced models with linear complexity, *e.g.*, Mamba2, RWKV6, Gated Linear Attention, *etc*, and identify two key features—attention normalization and non-causal inference—that enhance high-resolution visual generation performance. Building on these insights, we introduce a generalized linear attention paradigm, which serves as a low-rank approximation of a wide spectrum of popular linear token mixers. To save the training cost and better leverage pre-trained models, we initialize our models and distill the knowledge from pre-trained StableDiffusion (SD). We find that the distilled model, termed LinFusion, achieves performance on par with or superior to the original SD after only modest training, while significantly reducing time and memory complexity. Extensive experiments on SD-v1.5, SD-v2.1, and SD-XL demonstrate that LinFusion enables satisfactory and efficient zero-shot cross-resolution generation, accommodating ultra-resolution images like 16K on a single GPU. Moreover, it is highly compatible with pre-trained SD components and pipelines, such as ControlNet, IP-Adapter, DemoFusion, DistriFusion, *etc*, requiring no adaptation efforts.

## 1 INTRODUCTION

Recent years have witnessed significant advancements in AI-generated content (AIGC) with diffusion models Croitoru et al. (2023); Yang et al. (2023a). On the one hand, unlike classic models like

GAN Goodfellow et al. (2014), diffusion models refine noise vectors iteratively to produce high-quality results with fine details Nichol & Dhariwal (2021); Dhariwal & Nichol (2021); Rombach et al. (2022); Ho et al. (2020). On the other hand, having trained on large-scale data pairs, these models exhibit satisfactory alignment between input conditions and output results. These capabilities have spurred recent advancements in text-to-image generation Balaji et al. (2022); Ding et al. (2022); Nichol et al. (2021); Ramesh et al. (2022); Betker et al. (2023); Rombach et al. (2022); Saharia et al. (2022). Benefiting from the impressive performance and the open-source community, Stable Diffusion (SD) Rombach et al. (2022) stands out as one of the most popular models.

The success of models like SD can be largely attributed to their robust backbone structures for denoising. From UNet architectures with attention layers Ronneberger et al. (2015); Rombach et al. (2022) to Vision Transformers Peebles & Xie (2023); Bao et al. (2023); Chen et al. (2023); Esser et al. (2024), existing designs rely heavily on self-attention mechanisms to manage complex relationships between spatial tokens. Despite their impressive performance, the quadratic time and memory complexity inherent in self-attention operations poses significant challenges for high-resolution visual generation. For instance, as illustrated in Fig. 2(a), using FP16 precision, SD-v1.5 fails to generate 2048-resolution images on A100, a GPU with 80GB of memory, due to out-of-memory errors, making higher resolutions or larger models even more problematic[1].

To address these issues, in this paper, we aim at a novel token-mixing mechanism with linear complexity to the number of spatial tokens, offering an alternative to the classic self-attention approach. Inspired by recently introduced models with linear complexity, such as Mamba Gu & Dao (2023) and Mamba2 Dao & Gu (2024), which have demonstrated significant potential in sequential generation tasks, we first investigate their applicability as token mixers in diffusion models.

However, there are two drawbacks of Mamba diffusion models. On the one hand, when a diffusion model operates at a resolution different from its training scale, our theoretical analysis reveals that the feature distribution tends to shift, leading to difficulties in cross-resolution inference. On the other hand, diffusion models perform a denoising task rather than an auto-regressive task, allowing the model to simultaneously access all noisy spatial tokens and generate denoised tokens based on the entire input. In contrast, Mamba is fundamentally an RNN that processes tokens sequentially, meaning that the generated tokens are conditioned only on preceding tokens, a constraint termed causal restriction. Applying Mamba directly to diffusion models would impose this unnecessary causal restriction on the denoising process, which is both unwarranted and counterproductive. Although bi-directional scanning branches can somewhat alleviate this issue, the problem inevitably persists within each branch.

Focusing on the above drawbacks of Mamba for diffusion models, we propose a generalized linear attention paradigm. Firstly, to tackle the distribution shift between training resolution and larger inference resolution, a normalizer for Mamba, defined by the cumulative impact of all tokens on the current token, is devised to the aggregated features, ensuring that the total impact remains consistent regardless of the input scale. Secondly, we aim at a non-causal version of Mamba. We start our exploration by simply removing the lower triangular causal mask applied on the forget gate and find that all tokens would end up with identical hidden states, which undermines the model's capacity. To address this issue, we introduce distinct groups of forget gates for different tokens and propose an efficient low-rank approximation, enabling the model to be elegantly implemented in a linear-attention form. We analyze the proposed approach technically alongside recently introduced linear-complexity token mixers such as Mamba2 Dao & Gu (2024), RWKV6 Peng et al. (2024), and Gated Linear Attention Yang et al. (2023b) and reveal that our model can be regarded as a generalized non-causal version of these popular models.

The proposed generalized linear attention module is integrated into the architectures of SD, replacing the original self-attention layers, and the resultant model is termed as Linear-Complexity Diffusion Model, or *LinFusion* in short. By only training the linear attention modules for 50k iterations in a knowledge distillation framework, LinFusion achieves performance on par with or even superior to the original SD, while significantly reducing time and memory complexity, as shown in Fig. 2. Meanwhile, it delivers satisfactory zero-shot cross-resolution generation performance and can generate images at 16K resolution on a single GPU. It is also compatible with existing components and

---

[1]PyTorch 1.13 is adopted here for evaluation to reflect the theoretical complexity of various architectures. On higher versions of PyTorch, block-wise strategies are applied for memory efficient attention. However, the time efficiency is still a problem. Please refer to the appendix for more discussions on efficient implementations.

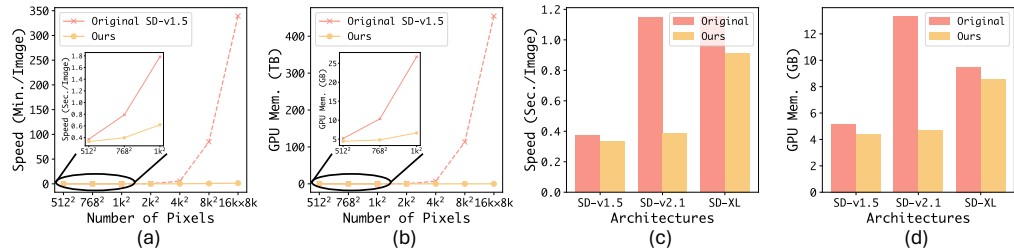

Figure 2: (a) and (b): Comparisons of the proposed LinFusion with original SD-v1.5 under various resolutions in terms of generation speed using 8 steps and GPU memory consumption. The dashed lines denote estimated values using quadratic functions due to out-of-memory error. (c) and (d): Efficiency comparisons on various architectures under their default resolutions.

pipelines for SD, such as ControlNet Zhang et al. (2023), IP-Adapter Ye et al. (2023), DemoFusion Du et al. (2024), DistriFusion Li et al. (2024), *etc*, allowing users to achieve various purposes with the proposed LinFusion flexibly without any additional training cost. As shown in Fig. 1, extensive experiments on SD-v1.5, SD-v2.1, and SD-XL validate the effectiveness of the proposed model and method. Our contributions can be summarized as follows:

- We investigate the non-causal and normalization-aware version of Mamba and propose a novel linear attention mechanism that addresses the challenges of high-resolution visual generation with diffusion models.

- Our theoretical analysis indicates that the proposed model is technically a generalized and efficient low-rank approximation of existing popular linear-complexity token mixers.

- Extensive experiments on SD demonstrate that the proposed LinFusion can achieve even better results than the original SD and exerts satisfactory zero-shot cross-resolution generation performance and compatibility with existing components and pipelines for SD. To the best of our knowledge, this is the first exploration of linear-complexity token mixers on the SD series model for text-to-image generation.

## 2 METHODOLOGY

### 2.1 PRELIMINARY

**Diffusion Models.** As a popular model for text-to-image generation, Stable Diffusion Rombach et al. (2022) (SD) first learns an auto-encoder $(\mathcal{E}, \mathcal{D})$, where the encoder $\mathcal{E}$ maps an image $x$ to a lower dimensional latent space: $z \leftarrow \mathcal{E}(x)$, and the decoder $\mathcal{D}$ learns to decode $z$ back to the image space $\hat{x} \leftarrow \mathcal{D}(z)$ such that $\hat{x}$ is close to the original $x$. In the inference time, a Gaussian noise in the latent space $z_T$ is sampled randomly and denoised by a UNet $\epsilon_\theta$ for $T$ steps. The denoised latent code after the final step $z_0$ is decoded by $\mathcal{D}$ to derive a generated image. In training, given an image $x$ and its corresponding text description $y$, $\mathcal{E}$ is utilized to obtain its corresponding latent code, and we add a random Gaussian noise $\epsilon$ for its noisy version $z_t$ with respect to the $t$-th step. The UNet is trained via the noise prediction loss $\mathcal{L}_{simple}$ Ho et al. (2020); Nichol & Dhariwal (2021):

$$\theta = \arg\min_\theta \mathbb{E}_{z \sim \mathcal{E}(x), y, \epsilon \sim \mathcal{N}(0,1), t}[\mathcal{L}_{simple}] \quad \mathcal{L}_{simple} = \|\epsilon - \epsilon_\theta(z_t, t, y)\|_2^2. \quad (1)$$

The UNet in SD contains multiple self-attention layers as token mixers to handle spatial-wise relationships and multiple cross-attention layers to handle text-image relationships. Given an input feature map in the UNet backbone $X \in \mathbb{R}^{n \times d}$ and weight parameters $W_Q, W_K \in \mathbb{R}^{d \times d'}$ and $W_V \in \mathbb{R}^{d \times d}$, where $n$ is the number of spatial tokens, $d$ is the feature dimension, and $d'$ is the attention dimension, self-attention can be formalized as:

$$Y = MV, \quad M = \text{softmax}(QK^\top / \sqrt{d'}), \quad Q = XW_Q, \quad K = XW_K, \quad V = XW_V. \quad (2)$$

We can observe from Eq. 2 that the complexity of self-attention is quadratic with respect to $n$ since the attention matrix $M \in \mathbb{R}^{n \times n}$, we mainly focus on its alternatives in this paper and are dedicated on a novel module for token mixing with linear complexity.

**Mamba.** As an alternative to Transformer Vaswani et al. (2017), Mamba Gu & Dao (2023) is proposed to handle sequential tasks with linear complexity with respect to the sequence length. At the heart of Mamba lies the State Space Model (SSM), which can be written as:

$$H_i = A_i \odot H_{i-1} + B_i^\top X_i = \sum_{j=1}^{i} \{ (\prod_{k=j+1}^{i} A_k) \odot (B_j^\top X_j) \}, \quad Y_i = C_i H_i, \tag{3}$$

where $i$ is the index of the current token in a sequence, $H_i$ denotes the hidden state, $X_i$ and $Y_i$ are row vectors denoting the $i$-th rows of the input and output matrices respectively, $A_i$, $B_i$, and $C_i$ are input-dependent variables, and $\odot$ indicates element-wise multiplication.

## 2.2 OVERVIEW

In the latest version, *i.e.*, Mamba2 Dao & Gu (2024), $A_i$ is a scalar, $B_i, C_i \in \mathbb{R}^{1 \times d'}$, $X_i, Y_i \in \mathbb{R}^{1 \times d}$, and $H_i \in \mathbb{R}^{d' \times d}$. According to State-Space Duality (SSD), the computation in Eq. 3 can be reformulated as the following expression, referred to as 1-Semiseparable Structured Masked Attention:

$$Y = ((CB^\top) \odot \tilde{A})X, \tag{4}$$

where $\tilde{A}$ is a $n \times n$ lower triangular matrix and $\tilde{A}_{ij} = \prod_{k=j+1}^{i} A_k$ for $j \leq i$. Such a matrix $\tilde{A}$ is known as 1-semiseparable, ensuring that Mamba2 can be implemented with linear complexity in $n$.

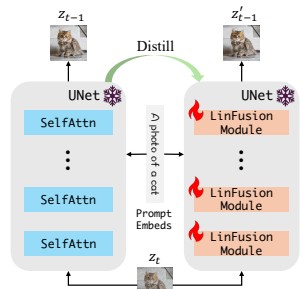

Figure 3: Overview of Lin-Fusion. We replace self-attention layers in the original SD with our LinFusion modules and adopt knowledge distillation to optimize the parameters.

In this paper, we aim at a diffusion backbone for the general text-to-image problems with linear complexity with respect to the number of image pixels. To this end, instead of training a novel model from scratch, we initialize and distill the model from pre-trained SD. Specifically, we utilize the SD-v1.5 model by default and substitute its self-attention—the primary source of quadratic complexity—with our proposed LinFusion modules. Only the parameters in these modules are trainable, while the rest of the model remains frozen. We distill knowledge from the original SD model into LinFusion such that given the same inputs, their outputs are as close as possible. Fig. 3 provides an overview of this streamline.

This approach offers two key benefits: (1) Training difficulty and computational overhead are significantly reduced, as the student model only needs to learn spatial relationships, without the added complexity of handling other aspects like text-image alignment; (2) The resulting model is highly compatible with existing components trained on the original SD models and their fine-tuned variations, since we only replace the self-attention layers with LinFusion modules, which are trained to be functionally similar to the original ones while maintaining the overall architecture.

Technically, to derive a linear-complexity diffusion backbone, one simple solution is to replace all the self-attention blocks with Mamba2, as shown in Fig. 4(a). We apply bi-directional SSM to ensure that the current position can access information from subsequent positions. Moreover, the self-attention modules in Stable Diffusion do not incorporate gated operations Hochreiter & Schmidhuber (1997); Cho (2014) or RMS-Norm Zhang & Sennrich (2019) as used in Mamba2. As shown in Fig. 4(b), we remove these structures to maintain the consistency and result in a slight improvement in performance. In the following parts of this section, we delve into the issues of applying SSM, the core module in Mamba2, to diffusion models and accordingly introduce the key features in LinFusion: normalization and non-causality in Secs. 2.3 and 2.4 respectively. Finally, in Sec. 2.5, we provide the training objectives to optimize parameters in LinFusion modules.

## 2.3 NORMALIZATION-AWARE MAMBA

In practice, we find that SSM-based structure shown in Fig. 4(b) can achieve satisfactory performance if the training and inference have consistent image resolutions. However, it fails when their image scales are different. We refer readers to Sec. 3.2 for the experimental results. To identify the cause of this failure, we examine the channel-wise means of the input and output feature maps, which exhibit the following proposition:

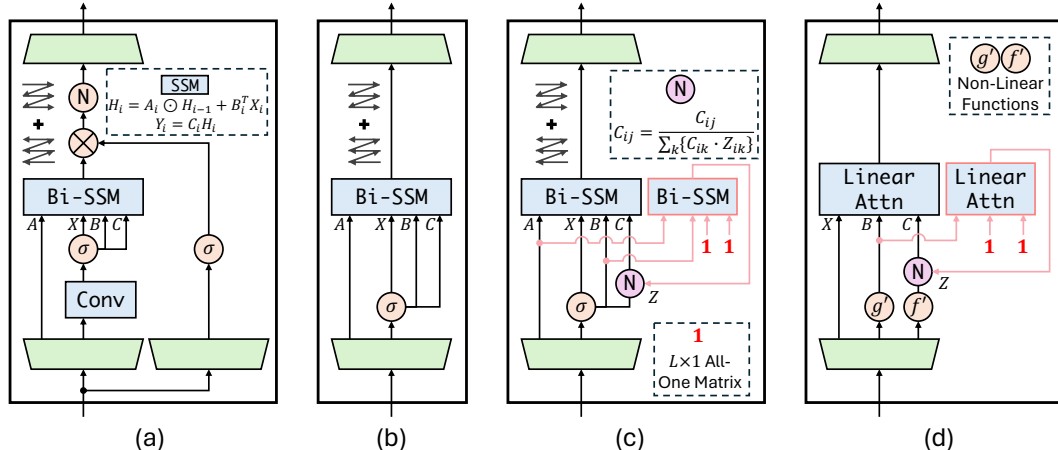

Figure 4: (a) The architecture of Mamba2. Bi-directional SSM is additionally involved here. (b) Mamba2 without gating and RMS-Norm. (c) Normalization-aware Mamba2. (d) The proposed LinFusion module with generalized linear attention.

**Proposition 1.** *Assuming that the mean of the $j$-th channel in the input feature map $X$ is $\mu_j$, and denoting $(CB^\top) \odot \tilde{A}$ as $M$, the mean of this channel in the output feature map $Y$ is $\mu_j \sum_{k=1}^{n} M_{ik}$.*

The proof is straightforward. We observe through Fig. 4(b) that there is non-negative activation applied on $X$, $B$, and $C$. Given that $A$ is also non-negative in Mamba2, according to Prop. 1, the channel-wise distributions would shift if $n$ is inconsistent in training and inference, which further leads to distorted results.

Solving this problem requires unifying the impact of all tokens on each one to the same scale, a property inherently provided by the `Softmax` function. In light of this, we propose normalization-aware Mamba in this paper, enforcing that the sum of attention weights from each token equals 1, *i.e.*, $\sum_{k=1}^{n} M_{ik} = 1$, which is equivalent to applying the SSM module one more time to obtain the normalization factor $Z$:

$$Z_i = A_i \odot Z_{i-1} + B_i, \quad C'_{ij} = \frac{C_{ij}}{\sum_{k=1}^{d'} \{C_{ik} \odot Z_{ik}\}}. \tag{5}$$

The operations are illustrated in Fig. 4(c). Experiments indicate that such normalization substantially improve the performance of zero-shot cross-resolution generalization.

## 2.4 Non-Causal Mamba

While bi-directional scanning enables a token to receive information from subsequent tokens—a crucial feature for diffusion backbones—treating feature maps as 1D sequences compromises the intrinsic spatial structures in 2D images and higher-dimensional visual content. To address this dilemma more effectively, we focus on developing a non-causal version of Mamba in this paper.

Non-causality indicates that one token can access to all tokens for information mixing, which can be achieved by simply removing the lower triangular causal mask applied on $\tilde{A}$. Thus, the recursive formula in Eq. 3 would become $H_i = \sum_{j=1}^{n} \{(\prod_{k=j+1}^{n} A_k) \odot (B_j^\top X_j)\}$. We observe that $H_i$ remains invariant with respect to $i$ in this formula. This implies that the hidden states of all tokens are uniform, which fundamentally undermines the intended purpose of the forget gate $A$. To address this issue, we associate different groups of $A$ to various input tokens. In this case, $A$ is a $n \times n$ matrix and $H_i = \sum_{j=1}^{n} \{(\prod_{k=j+1}^{n} A_{ik}) \odot (B_j^\top X_j)\}$. The $\tilde{A}_{ij}$ in Eq. 4 becomes $\prod_{k=j+1}^{n} A_{ik}$. Compared with that in Eq. 4, $\tilde{A}$ here is not necessarily 1-semiseparable. To maintain linear complexity, we impose the assumption that $\tilde{A}$ is low-rank separable, *i.e.*, there exist input-dependent matrices $F$ and $G$ such that $\tilde{A} = FG^\top$. In this way, the following proposition ensures that Eq. 4 under this circumstance can be implemented via linear attention:

**Proposition 2.** *Given that $\tilde{A} = FG^\top$, $F, G \in \mathbb{R}^{n \times r}$, and $B, C \in \mathbb{R}^{n \times d'}$, denoting $C_i = c(X_i)$, $B_i = b(X_i)$, $F_i = f(X_i)$, and $G_i = g(X_i)$, there exist corresponding functions $f'$ and $g'$ such that Eq. 4 can be equivalently implemented as linear attention, expressed as $Y = f'(X)g'(X)^\top X$.*

The proof can be found in the appendix. In practice, we adopt two MLPs to mimic the functionalities of $f'$ and $g'$. Combining with the normalization operations mentioned in Sec. 2.3, we derive an elegant structure shown in Fig. 4(d).

Not only that, we further demonstrate that the form of linear attention described in Proposition 2 can be extended to the more general case where $\tilde{A}_{ij}$ is a $d'$-dimension vector rather than a scalar:

**Proposition 3.** *Given that $\tilde{A} \in \mathbb{R}^{d' \times n \times n}$, if for each $1 \le u \le d'$, $\tilde{A}_u$ is low-rank separable: $\tilde{A}_u = F_u G_u^\top$, where $F_u, G_u \in \mathbb{R}^{n \times r}$, $F_{uiv} = f(X_i)_{uv}$, and $G_{ujv} = g(X_j)_{uv}$, there exist corresponding functions $f'$ and $g'$ such that the computation $Y_i = C_i H_i = C_i \sum_{j=1}^n \{\tilde{A}_{:ij} \odot (B_j^\top X_j)\}$ can be equivalently implemented as linear attention, expressed as $Y_i = f'(X_i)g'(X)^\top X$, where $\tilde{A}_{:ij}$ is a column vector and can broadcast to a $d' \times d$ matrix.*

The proof is provided in the appendix. From this point of view, the proposed structure can be deemed as a generalized linear attention and a non-causal form of recent linear-complexity sequential models, including Mamba2 Dao & Gu (2024), RWKV6 Peng et al. (2024), GLA Yang et al. (2023b), *etc*. In Tab. 1, we provide a summary of the parameterization in recent works for $A_i$.

## 2.5 TRAINING OBJECTIVES

In this paper, we replace all self-attention layers in the original SD with LinFusion modules. Only the parameters within these modules are trained, while all others remain frozen. To ensure that LinFusion closely mimics the original functionality of self-attention, we augment the standard noise prediction loss $\mathcal{L}_{simple}$ in Eq. 1 with additional

| Model | Parameterization of $A_i$ | Causal |
|---|---|---|
| Mamba2 Dao & Gu (2024) | $A_i \in \mathbb{R}$ | Yes |
| mLSTM Beck et al. (2024); Peng et al. (2021) | $A_i \in \mathbb{R}$ | Yes |
| Gated Retention Sun et al. (2024) | $A_i \in \mathbb{R}$ | Yes |
| GateLoop Katsch (2023) | $A_i \in \mathbb{R}^{d'}$ | Yes |
| HGRN2 Qin et al. (2024) | $A_i \in \mathbb{R}^{d'}$ | Yes |
| RWKV6 Peng et al. (2024) | $A_i \in \mathbb{R}^{d'}$ | Yes |
| Gated Linear Attention Yang et al. (2023b) | $A_i \in \mathbb{R}^{d'}$ | Yes |
| MLLA Han et al. (2024) | $A_{ij} = 1$ | No |
| VSSD Shi et al. (2024) | $A_{ij} \in \mathbb{R}$ | No |
| Generalized Linear Attention | $\tilde{A}_{ij} \in \mathbb{R}^{d'}$ | No |

Table 1: A summary of the parameterization in recent linear token mixers for $A_i$, partially adapted from Yang et al. (2023b).

losses. Specifically, we introduce a knowledge distillation loss $\mathcal{L}_{kd}$ to align the final outputs of the student and teacher models and a feature matching loss $\mathcal{L}_{feat}$ to match the outputs of each LinFusion module and the corresponding self-attention layer. The training objectives can be written as:

$$\theta = \arg\min_\theta \mathbb{E}_{z \sim \mathcal{E}(x), y, \epsilon \sim \mathcal{N}(0,1), t}[\mathcal{L}_{simple} + \alpha\mathcal{L}_{kd} + \beta\mathcal{L}_{feat}],$$

$$\mathcal{L}_{kd} = \|\epsilon_\theta(z_t, t, y) - \epsilon_{\theta_{org}}(z_t, t, y)\|_2^2, \quad \mathcal{L}_{feat} = \frac{1}{L}\sum_{l=1}^L \|\epsilon_\theta^{(l)}(z_t, t, y) - \epsilon_{\theta_{org}}^{(l)}(z_t, t, y)\|_2^2, \quad (6)$$

where $\alpha$ and $\beta$ are hyper-parameters controlling the weights of the respective loss terms, $\theta_{org}$ represents parameters of the original SD, $L$ is the number of LinFusion/self-attention modules, and the superscript $^{(l)}$ refers to the output of the $l$-th one in the diffusion backbone.

## 3 EXPERIMENTS

### 3.1 IMPLEMENTATION DETAILS

We present qualitative results on SD-v1.5, SD-v2.1, and SD-XL in Fig. 5 and mainly conduct experiments on SD-v1.5 in this section. There are 16 self-attention layers in SD-v1.5 and we replace them with LinFusion modules proposed in this paper. Functions $f'$ and $g'$ mentioned in Proposition 2 are implemented as MLP, which consists of a linear branch and a non-linear branch with one `Linear-LayerNorm-LeakyReLU` block. The number of newly introduced parameters by them is less than 6% and 1% of UNets in SD-v1.5 and SD-XL, respectively. Their results are added to form the outputs of $f'$ and $g'$. The parameters of the linear branch in $f'$ and $g'$ are initialized as $W_Q$ and $W_K$ respectively, while the outputs of the non-linear branch are initialized as $0$. We use only 169k images in LAION Schuhmann et al. (2022) with aesthetics scores larger than $6.5$ for training and adopt the BLIP2 Li et al. (2023) image captioning model to regenerate the textual descriptions, which is significantly less than the amount of data required for training the original text-to-image

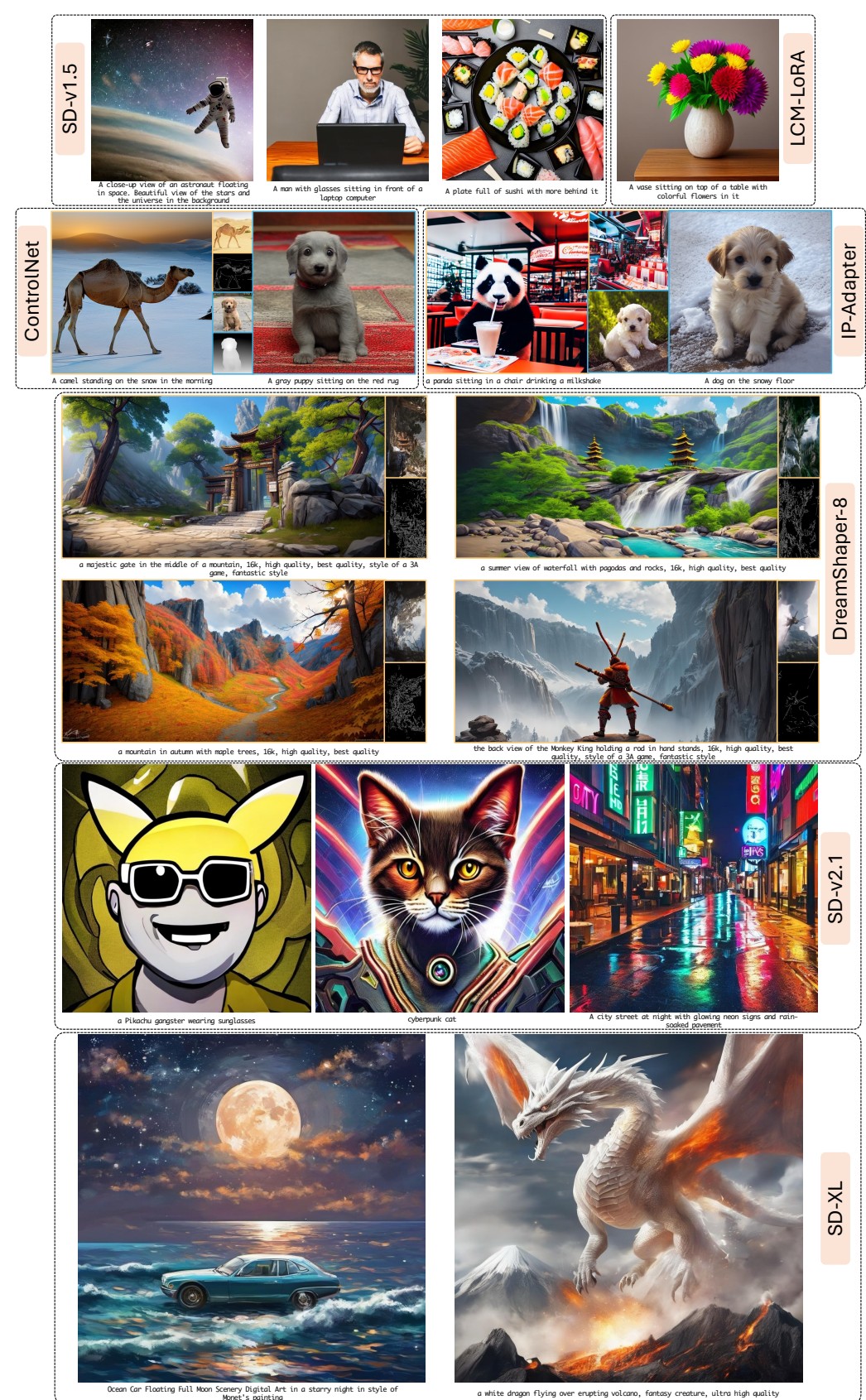

Figure 5: Qualitative text-to-image results by LinFusion based on various architectures.

| ID | Setting | FID($\downarrow$) | CLIP-T($\uparrow$) | GPU Memory (GB) | Time (sec./image) |
|----|---------|-----|--------|-----------------|-------------------|
| A | Original SD (v1.5) | 12.86 | 0.321 | 5.17 | 2.32 |
| B | Distilled Diffusion Model (Base) Kim et al. (2023a) | 16.63 | 0.315 | 4.62 | 1.58 |
| C | Distilled Diffusion Model (Small) Kim et al. (2023a) | 18.58 | 0.297 | 4.45 | 1.44 |
| D | Distilled Diffusion Model (Tiny) Kim et al. (2023a) | 18.82 | 0.295 | 4.13 | 1.32 |
| E | EfficientViT Cai et al. (2023) | 17.54 | 0.310 | 4.62 | 4.33 |
| F | DiG Zhu et al. (2024a) | 17.51 | 0.309 | 4.86 | 2.41 |
| G | Vision Mamba Zhu et al. (2024b) | 18.36 | 0.307 | 4.80 | 4.18 |
| H | Bi-Directional Mamba2 | 18.90 | 0.307 | 4.70 | 4.54 |
| I | H - Gating - RMS Norm | 17.30 | 0.309 | 4.69 | 4.33 |
| J | I + Normalization | 17.60 | 0.308 | 4.73 | 6.51 |
| K | J - SSM + Linear Attn. | 17.63 | 0.307 | 4.09 | 2.07 |
| L | J - SSM + Generalized Linear Attn. | 17.07 | 0.309 | 4.43 | 2.07 |
| M | L + $\mathcal{L}_{kd}$ + $\mathcal{L}_{feat}$ | **12.57** | **0.323** | 4.43 | 2.07 |

Table 2: Performance and efficiency comparisons with various baselines on the COCO benchmark.

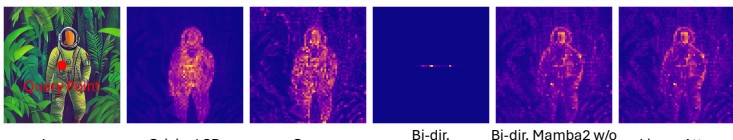

Figure 6: Visualization of attention maps by various architectures. The prompt is "`Astronaut in a jungle, cold color palette, muted colors, detailed, 8k`".

models. Both hyper-parameters, $\alpha$ and $\beta$, are set as 0.5, following the approach taken in Kim et al. (2023a), which also focuses on the architectural distillation of SD. The model is optimized using AdamW Loshchilov & Hutter (2017) with a learning rate of $10^{-4}$. Training is conducted on 8 RTX6000Ada GPUs with a total batch size of 96 under $512 \times 512$ resolution for 100k iterations, requiring $\sim 1$ day to complete. The efficiency evaluations are conducted on a single NVIDIA A100-SXM4-80GB GPU.

## 3.2 MAIN RESULTS

**Ablation Studies.** To demonstrate the effectiveness of the proposed LinFusion, we report the comparison results with alternative solutions such as those shown in Fig. 4(a), (b) and (c). We follow the convention in previous works focusing on text-to-image generation Kang et al. (2023) and conduct a quantitative evaluation on the COCO benchmark Lin et al. (2014) containing 30k text prompts. The metrics are FID Heusel et al. (2017) against the COCO2014 test dataset and the cosine similarity in the CLIP-ViT-G feature space Radford et al. (2021). We also report the running time per image with 50 denoising steps and the GPU memory consumption during inference for efficiency comparisons. Results under $512 \times 512$ resolution are shown in Tab. 2.

**Mitigating Structural Difference.** We begin our exploration from the original Mamba2 structure Dao & Gu (2024) with bi-directional scanning, *i.e.*, Fig. 4(a), and try removing the gating and RMS-Norm, *i.e.*, Fig. 4(b), to maintain a consistent holistic structure with the self-attention layer in the original SD. In this way, the only difference with the original SD lies in the SSM or self-attention for token mixing. We observe that such structural alignment is beneficial for performance.

**Normalization and Non-Causality.** We then apply the proposed normalization operation and the non-causal treatment sequentially, corresponding to Fig. 4(c) and (d). Although results in Tab. 2 indicate that normalization would slightly hurt the performance, we will show in the following Tab. 3 that it is crucial for generating images with resolutions unseen during training. Further adding the proposed non-causal treatment, we obtain results better than Fig. 4(b).

We also compare the proposed non-causal operation with the simplified case mentioned in Sec. 2.4, achieved by directly removing the lower triangular causal mask applied on $\tilde{A}$, which results in a 1-rank matrix, *i.e.*, various tokens share the same group of forget gates. The inferior results demonstrate the effectiveness of the proposed generalized linear attention.

**Attention Visualization.** In Fig. 6, we visualize the self-attention maps yielded by various methods, including the original SD, bi-directional SSM, linear attention with shared forget gates, and generalized linear attention in LinFusion. Results indicate that our method works better for capturing a broader range of spatial dependency and best matches the predictions of the original SD.

**Knowledge Distillation and Feature Matching.** We finally apply loss terms $\mathcal{L}_{kd}$ and $\mathcal{L}_{feat}$ in Eq. 6, which enhance the performance further and even surpass the SD teacher.

| Setting | FID($\downarrow$) | CLIP-T($\uparrow$) |
|---|---|---|
| Original SD (v1.5) | 32.71 | 0.290 |
| Bi-Directional Mamba2 | 196.72 | 0.080 |
| +Normalization | 37.02 | 0.273 |
| Bi-Directional Mamba2 w/o Gating & RMS-Norm | 134.78 | 0.158 |
| +Normalization | 50.30 | 0.263 |
| Generalized Linear Attention | 359.64 | 0.069 |
| +Normalization | **36.33** | **0.285** |

Table 3: Normalization is crucial for cross-resolution generation as demonstrated by the results on the COCO benchmark under $1024 \times 1024$ resolution, which is unseen in training.

Figure 7: Qualitative studies of normalization on various architectures. The resolution is $4096 \times 512$ and the prompt is "A group of golden retriever puppies playing in snow. Their heads pop out of the snow covered in".

**Cross-Resolution Inference.** It is desirable for diffusion model to generate images of unseen resolutions during training–a feature of the original SD. Since modules other than LinFusion are pre-trained and fixed in our work, normalization is a key component for this feature to maintain consistent feature distributions for training and inference. We report the results of $1024 \times 1024$ resolution in Tab. 3, which indicate that the conclusion holds for all the basic structures such as Mamba2, Mamba2 without gating and RMS-Norm, and the proposed generalized linear attention. Fig. 7 shows a qualitative example, where results without normalization are meaningless.

## 3.3 EMPIRICAL EXTENSIONS

The proposed LinFusion is highly compatible with various components/pipelines for SD, such as ControlNet Zhang et al. (2023), IP-Adapter Ye et al. (2023), LoRA Hu et al. (2022), DemoFusion Du et al. (2024), DistriFusion Li et al. (2024), *etc*, without any further training or adaptation. We present some qualitative results in Fig. 5 and refer readers to the appendix for more results. The overall performance of LinFusion is comparable with the original SD.

**ControlNet.** ControlNet Zhang et al. (2023) introduces plug-and-play components to SD for additional conditions, such as edge, depth, and semantic map. We substitute SD with the proposed LinFusion and compare the FID, CLIP score, and the similarity between the input conditions and the extracted conditions from generated images of diffusion models with the original SD. The results are shown in Tab. 4.

**IP-Adapter.** Personalized text-to-image generation Gal et al. (2022) is a popular application of SD, which focuses on generating images simultaneously following both input identities and textual descriptions. IP-Adapter Ye et al. (2023) offers a zero-shot solution that trains a mapper from the image space to the condition space of SD so that it can handle both image and text conditions. We demonstrate that IP-Adapter trained on SD can be used directly on LinFusion. The performance on the DreamBooth dataset Ruiz et al. (2023), containing 30 identities and 25 text prompts to form 750 test cases in total, is shown in Tab. 6. We use 5 random seeds for each case and report the averaged CLIP image similarity, DINO Caron et al. (2021) image similarity, and CLIP text similarity.

**LoRA.** Low-rank adapters (LoRA) Hu et al. (2022) aim at low-rank matrices applied on the weights of a basic model such that they can be adapted for different tasks or purposes. For instance, Luo et al. (2023b) introduce LCM-LoRA such that the pre-trained SD can be used for LCM inference with only a few denoising steps Luo et al. (2023a). Here, we directly apply LoRA in the LCM-LoRA model to LinFusion. The performance on the COCO benchmark is shown in Tab. 7. Since LCM adopts different training objectives with the original diffusion model, the generalization performance measured by FID is relatively worse in this case compared to other settings.

**Ultrahigh-Resolution Generation.** As discussed in Huang et al. (2024); He et al. (2024), directly applying diffusion models trained on low resolutions for higher-resolution generation can result in content distortion and duplication. A series of works are dedicated to higher-resolution image generation by leveraging off-the-shelf diffusion models Du et al. (2024); Lin et al. (2024a;b); Haji-Ali et al. (2024). However, limited by the quadratic-complexity self-attention, when applied for ultrahigh-resolution generation, existing approaches turn to patch-wise strategies to overcome the heavy computation burden Bar-Tal et al. (2023), which leads to inferior results, as shown in Settings

| Type | Canny Edge | | Depth | |
|---|---|---|---|---|
| Method | F1($\uparrow$) | CLIP-T($\uparrow$) | RMSE($\downarrow$) | CLIP-T($\uparrow$) |
| Original SD (v1.5) | 0.210 | 0.296 | **9.364** | **0.300** |
| LinFusion | **0.247** | **0.303** | 9.460 | 0.294 |

Table 4: Results of ControlNet on the original SD-v1.5 and LinFusion.

| ID | Setting | FID($\downarrow$) | CLIP-T($\uparrow$) | Time($\downarrow$) (sec.) |
|---|---|---|---|---|
| A | DemoFusion | 70.01 | 0.343 | 61.36 |
| B | A - Patch | 65.44 | 0.340 | 57.56 |
| C | B + SDEdit | 65.15 | **0.344** | 26.98 |
| D | C + LinFusion | **65.07** | 0.338 | **14.71** |

Table 5: Results of LinFusion on pipelines dedicated for high-resolution generation.

| Method | CLIP-T($\uparrow$) | CLIP-I($\uparrow$) | DINO($\uparrow$) |
|---|---|---|---|
| Original SD (v1.5) | **0.281** | 0.841 | 0.731 |
| LinFusion | 0.280 | **0.846** | **0.740** |

Table 6: Results of IP-Adapter on the original SD-v1.5 and LinFusion.

| Method | FID($\downarrow$) | CLIP-T($\uparrow$) |
|---|---|---|
| Original SD (v1.5) | **23.43** | **0.297** |
| LinFusion | 27.14 | 0.294 |

Table 7: Results of LCM-LoRA on the original SD-v1.5 and LinFusion.

| | Against Ground-Truth | | Against 1-GPU Results | | | Time($\downarrow$) | Speedup($\uparrow$) |
|---|---|---|---|---|---|---|---|
| Setting | LPIPS($\downarrow$) | FID($\downarrow$) | PSNR($\uparrow$) | LPIPS($\downarrow$) | FID($\downarrow$) | (sec.) | |
| SD-XL 1 GPU | 0.797 | **23.96** | - | - | - | 6.51 | - |
| w. LinFusion 1 GPU | **0.794** | 24.85 | - | - | - | **6.49** | - |
| DistriFusion 2 GPUs | 0.797 | **24.18** | 24.63 | 0.146 | 4.87 | 5.36 | 1.21 |
| w. LinFusion 2 GPUs | **0.795** | 24.96 | **26.45** | **0.113** | **4.09** | **3.85** | **1.69** |
| DistriFusion 4 GPUs | 0.798 | **24.22** | 23.05 | 0.183 | 5.77 | 4.22 | 1.54 |
| w. LinFusion 4 GPUs | **0.796** | 25.00 | **24.63** | **0.148** | **5.08** | **2.51** | **2.59** |
| DistriFusion 8 GPUs | 0.799 | **24.40** | 22.04 | 0.211 | **6.45** | 4.37 | 1.49 |
| w. LinFusion 8 GPUs | **0.797** | 24.97 | **22.93** | **0.198** | 6.61 | **2.14** | **3.03** |

Table 8: Results of distributed parallel inference on a server with 8 RTX 4090 D GPUs. Benefiting from its linear complexity and constant communication cost among various patches, LinFusion is readily for distributed parallel inference with multiple GPUs. Compared with DistriFusion, it achieves more significant acceleration even **without** NVLink.

A and B of Tab. 5. Note that removing patchification can be faster than the original implementation under $2048 \times 2048$ resolution here since it avoids looping over each image patch sequentially.

Complementary to these methods, LinFusion addresses the computational overhead via generalized linear attention. As shown in Settings C and D of Tab. 5, LinFusion achieves $\sim 2\times$ acceleration under $2048 \times 2048$ resolution. Instead of going through full denoising steps in the original Demo-Fusion Du et al. (2024), tricks in SDEdit Meng et al. (2021) are additionally applied here so that the former 60% steps are skipped, which further enhances the efficiency without scarifying the quality. Please refer to the appendix for more analysis. Backed up by the linear-complexity LinFusion, such strategies enable ultrahigh-resolution generation up to 16K on a single GPU as shown in Fig. 1.

**Distributed Parallel Inference.** LinFusion is friendly for distributed parallel inference benefiting from its linear complexity, given that the communication cost is constant with respect to image resolution. Specifically, unlike the original DistriFusion Li et al. (2024) requiring transmitting all the key and value tokens for self-attention communication, the transmission in LinFusion is $g'(X)^\top X \in \mathbb{R}^{c' \times c}$, which is not related with the number of image tokens. In consequence, as shown in Tab. 8, LinFusion does not require NVLink hardware to achieve satisfactory acceleration. Please refer to the appendix for qualitative examples.

## 4 CONCLUSION

This paper introduces a diffusion backbone termed LinFusion for text-to-image generation with linear complexity in the number of pixels. At the heart of LinFusion lies a generalized linear attention mechanism, distinguished by its normalization-aware and non-causal operations—key aspects overlooked by recent linear-complexity token mixers like Mamba, Mamba2, and GLA. We reveal theoretically that the proposed paradigm serves as a general low-rank approximation for the non-causal variants of recent models. Based on Stable Diffusion (SD), LinFusion modules after knowledge distillation can seamlessly replace self-attention layers in the original model, ensuring that LinFusion is highly compatible to existing components or pipelines for Stable Diffusion, like ControlNet, IP-Adapter, LoRA, DemoFusion, DistriFusion, *etc*, without any further training effort. Extensive experiments on SD-v1.5, SD-v2.1, and SD-XL demonstrate that the proposed model outperforms existing baselines and achieves performance on par with, or better than, the original SD with significantly reduced computational overhead. On a single GPU, it can accommodate image generation with resolutions up to 16K.

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

## A  RELATED WORKS

In this section, we review related works from two perspectives, namely efficient diffusion architectures and linear-complexity token mixers.

### A.1  EFFICIENT DIFFUSION ARCHITECTURES

There are mainly two mainstreams of works aiming at more efficient diffusion models, including efficient sampling for a reduced number of sampling time-steps Song et al. (2023); Luo et al. (2023a); Kim et al. (2023b); Ma et al. (2024b); Zhou et al. (2024) and efficient architectures for faster network inference. This paper focuses on the latter, which is a bottleneck for generating high-resolution visual results, particularly due to the self-attention token mixers in existing diffusion backbones.

To mitigate the efficiency issue triggered by the quadratic time and memory complexity, a series of works, including DiS Fei et al. (2024a), DiM Teng et al. (2024), DiG Zhu et al. (2024a), Diffusion-RWKV Fei et al. (2024b), DiffuSSM Yan et al. (2024), and Zigma Hu et al. (2024). These works have successfully adapted recent state space models like Mamba Gu & Dao (2023), RWKV Peng et al. (2023), or Linear Attention Katharopoulos et al. (2020) into diffusion architectures. However, these architectures maintain a causal restriction for diffusion tasks, processing input spatial tokens one by one, with generated tokens conditioned only on preceding tokens. In contrast, the diffusion task allows models to access all noisy tokens simultaneously, making the causal restriction unnecessary. To address this, we eliminate the causal restriction and introduce a non-causal token mixer specifically designed for the diffusion model.

Additionally, previous works have primarily focused on class-conditioned image generation. For text-to-image generation, Kim et al. (2023a) propose architectural pruning for Stable Diffusion (SD) by reducing the number of UNet stages and blocks, which is orthogonal to our focus on optimizing self-attention layers.

### A.2  LINEAR-COMPLEXITY TOKEN MIXERS

Despite the widespread adoption of Transformer Vaswani et al. (2017) across various fields due to its superior modeling capacity, the quadratic time and memory complexity of the self-attention mechanism often leads to efficiency issues in practice. A series of linear-complexity token mixers are thus introduced as alternatives, such as Linear Attention Katharopoulos et al. (2020), State Space Model Gu et al. (2021), and their variants including Mamba Gu & Dao (2023), Mamba2 Dao & Gu (2024), mLSTM Beck et al. (2024); Peng et al. (2021), Gated Retention Sun et al. (2024), DFW Mao (2022); Pramanik et al. (2023), GateLoop Katsch (2023), HGRN2 Qin et al. (2024), RWKV6 Peng et al. (2024), GLA Yang et al. (2023b), *etc*. These models are designed for tasks requiring sequential modeling, making it non-trivial to apply them to non-causal vision problems. Addressing this challenge is the main focus of our paper.

For visual processing tasks, beyond the direct treatment of inputs as sequences, there are concurrent works focused on non-causal token mixers with linear complexity. MLLA Han et al. (2024) employs Linear Attention Katharopoulos et al. (2020) as token mixers in vision backbones without a gating mechanism for hidden states. In VSSD Shi et al. (2024), various input tokens share the same group of gating values. In contrast, the model proposed in this paper relaxes these gating assumptions, offering a generalized non-causal version of various modern state-space models.

## B  THEORETICAL PROOF

**Proposition 1.** *Assuming that the mean of the $j$-th channel in the input feature map $X$ is $\mu_j$, and denoting $(CB^\top) \odot \tilde{A}$ as $M$, the mean of this channel in the output feature map $Y$ is $\mu_j \sum_{k=1}^{n} M_{ik}$.*

The proof is straightforward.

**Proposition 2.** *Given that $\tilde{A} = FG^\top$, $F, G \in \mathbb{R}^{n \times r}$, and $B, C \in \mathbb{R}^{n \times d'}$, denoting $C_i = c(X_i)$, $B_i = b(X_i)$, $F_i = f(X_i)$, and $G_i = g(X_i)$, there exist corresponding functions $f'$ and $g'$ such that Eq. 4 of the main manuscript can be equivalently implemented as linear attention, expressed as $Y = f'(X)g'(X)^\top X$.*

*Proof.* Given existing conditions, we have:

$$
\begin{aligned}
(CB^\top) \odot \tilde{A} &= [(c(X_i)b^\top(X_j)) \odot (f(X_i)g^\top(X_j))]_{i,j} \\
&= [(\sum_{u=1}^{d'}\{c(X_i)_u b(X_j)_u\})(\sum_{v=1}^{r}\{f(X_i)_v g(X_j)_v\})]_{i,j} \\
&= [\sum_{u=1}^{d'}\sum_{v=1}^{r}\{(c(X_i)_u f(X_i)_v)(b(X_j)_u g(X_j)_v)\}]_{i,j} \\
&= [(c(X_i) \otimes f(X_i))(b(X_j) \otimes g(X_j))^\top]_{i,j},
\end{aligned}
\tag{7}
$$

where $\otimes$ denotes Kronecker product. Defining $f'(X_i) = c(X_i) \otimes f(X_i)$ and $g'(X_i) = b(X_i) \otimes g(X_i)$, we derive $Y = f'(X)g'(X)^\top X$. □

**Proposition 3.** *Given that $\tilde{A} \in \mathbb{R}^{d' \times n \times n}$, if for each $1 \leq u \leq d'$, $\tilde{A}_u$ is low-rank separable: $\tilde{A}_u = F_u G_u^\top$, where $F_u, G_u \in \mathbb{R}^{n \times r}$, $F_{uiv} = f(X_i)_{uv}$, and $G_{ujv} = g(X_j)_{uv}$, there exist corresponding functions $f'$ and $g'$ such that the computation $Y_i = C_i H_i = C_i \sum_{j=1}^{n}\{\tilde{A}_{:ij} \odot (B_j^\top X_j)\}$ can be equivalently implemented as linear attention, expressed as $Y_i = f'(X_i)g'(X)^\top X$, where $\tilde{A}_{:ij}$ is a column vector and can broadcast to a $d' \times d$ matrix.*

*Proof.* Given existing conditions, we have:

$$
\begin{aligned}
Y_i &= \sum_{u=1}^{d'}[c(X_i)_u\{\sum_{j=1}^{n}\sum_{v=1}^{r}(f(X_i)_{uv}g(X_j)_{uv}b(X_j)_u X_j)\}] \\
&= \sum_{u=1}^{d'}\sum_{v=1}^{r}[c(X_i)_u f(X_i)_{uv}\sum_{j=1}^{n}\{g(X_j)_{uv}b(X_j)_u X_j\}] \\
&= \text{vec}(c(X_i) \cdot f(X_i))[\text{vec}(b(X_j) \cdot g(X_j))]_j^\top X,
\end{aligned}
\tag{8}
$$

where $f(X_i) = F_{:i:}$ and $g(X_j) = G_{:j:}$ are $d' \times r$ matrices, $\cdot$ denotes element-wise multiplication with broadcasting, and vec represents flatting a matrix into a row vector. Defining $f'(X_i) = \text{vec}(c(X_i) \cdot f(X_i))$ and $g'(X_i) = \text{vec}(b(X_j) \cdot g(X_j))$, we derive $Y = f'(X)g'(X)^\top X$. □

## C  ADDITIONAL EXPERIMENTS

**Ultrahigh-Resolution Generation.** We present qualitative examples to illustrate the effectiveness of LinFusion on ultrahigh-resolution generation in Fig. 8. We build LinFusion upon DemoFusion Du et al. (2024), a pipeline dedicated to high-resolution generation. Similar to SDEdit Meng et al. (2021), DemoFusion also generate high-resolution images in a coarse-to-fine fashion. In the original implementation, for efficiency, in the high-resolution upsampling stage, DemoFusion handles a high-resolution image patch-by-patch and averages the outputs of overlapped areas Bar-Tal et al. (2023). However, we find that such a patch-wise treatment largely ignores the holistic text-image relationships. As shown in Fig. 8(DemoFusion), there are stars on the body of the astronaut. With an efficient architecture introduced by LinFusion, we do not have to conduct inference patch-by-patch. Instead, the whole image, even in the ultra-high-resolution generation stage, can be accommodated to a single GPU for denoising, which addresses the above limitation effectively as shown in Fig. 8(Full Steps).

Moreover, DemoFusion has to conduct full steps in the high-resolution denoising stage, which would introduce significant latency. Motivated from the insight in SDEdit Meng et al. (2021) that early denoising steps tend to take over the overall image layouts, we propose to skip some initial steps in the high-resolution stage, given that the overall image structures have been produced in the low-resolution stage. We find that it not only improves the efficiency but also makes the pipeline more robust to the turbulence on image layout in the high-resolution stage, as shown in Fig. 8(40% Steps).

**Distributed Parallel Inference.** We supplement qualitative results of distributed parallel inference by building LinFusion upon DistriFusion Li et al. (2024) in Fig. 9. Using constant communica-

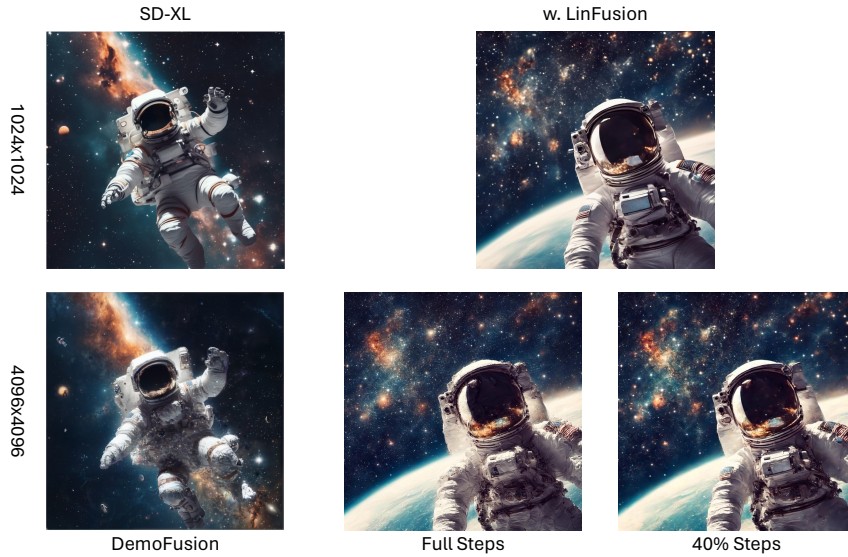

Figure 8: LinFusion is complementary to pipelines dedicated to high-resolution generation like DemoFusion. To enhance the performance, instead of working patch-by-patch, we handle the image as a whole benefiting from the efficient LinFusion architecture. Moreover, we reveal that skipping part of the denoising steps in the high-resolution stage can further improve the efficiency without hurting the performance. The prompt is "`An astronaut floating in space. Beautiful view of the stars and the universe in the background.`".

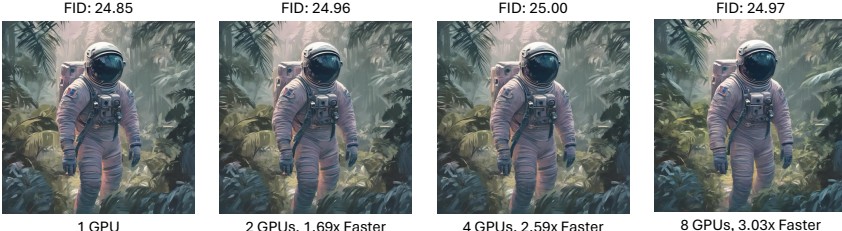

Figure 9: LinFusion is complementary to pipelines for high-resolution generation like DistriFusion. Using constant communication cost among various GPU, it achieves highly comparable performance with single-GPU inference. The prompt is "`Astronaut in a jungle, cold color palette, muted colors, detailed, 8k`".

| Setting | FID(↓) | CLIP-T(↑) |
|---|---|---|
| SD-v1.5 | 12.86 | 0.321 |
| w. LinFusion | **12.57** | **0.323** |
| SD-v2.1 | **12.84** | **0.333** |
| w. LinFusion | 13.84 | 0.329 |
| SD-XL | **14.74** | **0.340** |
| w. LinFusion | 15.72 | 0.338 |
| PixArt-Sigma | 26.32 | **0.334** |
| w. 75% LinFusion | **24.32** | 0.327 |

Table 9: Performance of LinFusion built upon various models on the COCO benchmark.

tion cost among various GPUs, it achieves highly comparable performance with single-GPU inference. Unlike the original DistriFusion, LinFusion offers significant acceleration using multiple GPUs without the dependency on NVLink.

**Results on More Architectures.** We conduct experiments on a variety of diffusion architectures in this paper, including SD-v1.5, SD-v2.1 Rombach et al. (2022), SD-XL Podell et al. (2023), and PixArt-Sigma Chen et al. (2023). The former three adopt transformer-based UNet while the last one is based on DiT Peebles & Xie (2022), a pure-Transformer structure. Their quantitative results are listed in Tab. 9.

We find that on SD-v2.1 and SD-XL, LinFusion leads to slightly inferior results. We speculate that the reason lies in the training data used for LinFusion, which consists of only ∼160K relatively

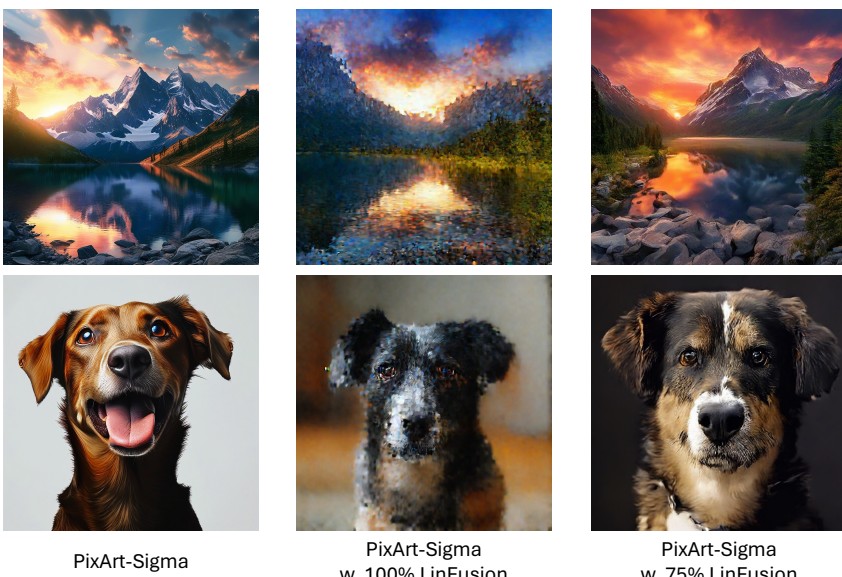

PixArt-Sigma      PixArt-Sigma
w. 100% LinFusion      PixArt-Sigma
w. 75% LinFusion

Figure 10: By default, LinFusion replaces all self-attention layers in a diffusion backbone. When applied to DiT-based structures like PixArt-Sigma, this configuration often struggles to generate smooth results. Leaving a small part of original self-attention layers unmoved, *e.g.*, 25%, could largely alleviate this challenge. The prompts are "`A photo of beautiful mountain with realistic sunset and blue lake, highly detailed, masterpiece`" and "`dog`" respectively.

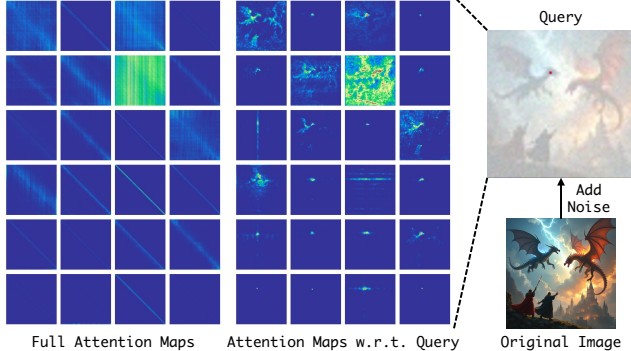

Full Attention Maps    Attention Maps w.r.t. Query    Original Image

Figure 11: Visualization of attention maps by various heads for an intermediate denoising step on FLUX-1.dev. Attention in pre-trained DiTs is not low-rank and is largely conducted in a local fashion.

low-resolution samples, the majority of which are below $512 \times 512$ resolution. Involving more high-quality samples can benefit the performance.

On PixArt-Sigma, we find that replacing all the self-attention layers in the DiT would result in unnatural results, as shown in Fig. 10. We speculate that the challenge arises because self-attention is the core and sole mechanism for managing token relationships in DiT. Replacing these layers entirely with LinFusion may create a significant divergence from the original architecture, leading to difficulties during training. As shown in Fig. 10, we leave a small part of the original self-attention layers unchanged, *e.g.*, 25% by evenly preserving 1 self-attention layer of every 4 layers, which could largely alleviate this challenge.

**Adaptation to MM-DiT.** Most state-of-the-art text-to-image models, like SD-3 Esser et al. (2024) and FLUX Labs (2024), adopt multi-model joint attention modules, which conduct self-attention operations on the concatenation of text and image tokens. For these models, we find that directly replacing all the joint attention layers with LinFusion modules may not produce reasonable images. We delve into the underlying reasons by visualizing the attention maps. As shown in Fig. 11, we

traditional Chinese street at night, red lanterns illuminating the cobblestone road, wooden storefronts with calligraphy signs, vendors selling snacks, soft mist in the air, ultra-detailed, photorealistic, ultra HD, 8K, vivid lighting, nostalgic atmosphere, intricate details of lanterns and signs

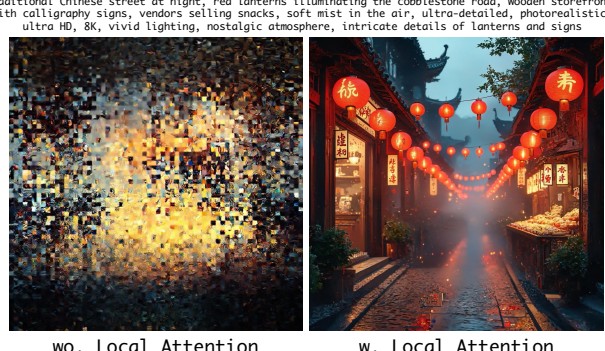

wo. Local Attention     w. Local Attention

Figure 12: On Diffusion Transformers based on multi-modal joint attention, *e.g.*, FLUX 1.dev, native LinFusion would generate meaningless results if all attention layers are replaced by linear attention. Local attention with a fixed window size, *e.g.*, 15, can largely alleviate the problem.

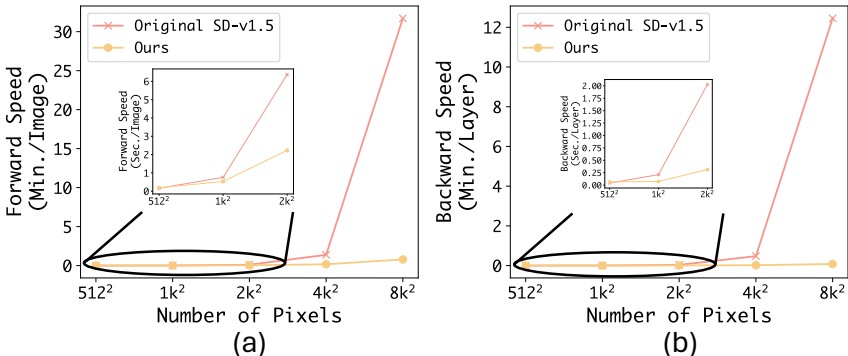

Figure 13: Comparisons of LinFusion with the original SD-v1.5 under various resolutions in terms of forward speed using 8 steps and backward speed over 1 layer, when FlashAttention 2 Dao (2023) is adopted for the original architecture and Triton implementation is applied for LinFusion.

find that the attention maps do not exhibit a low-rank property. As the core mechanism for token interaction, linear attention solely is inherently incapable of mimicking the functionalities of the vanilla attention mechanism.

Fortunately, we also find in Fig. 11 that most attention interactions demonstrate local patterns: tokens tend to aggregate information more from local neighborhoods. We thus augment the native LinFusion with local attention mechanisms similar to Hassani et al. (2023). In this way, local interactions can be handled the local attention layers effectively, while global interactions are processed by linear attention. Since the local window size would not vary with the increasing of image resolutions, this hybrid model is still linear-complexity. As shown in Fig. 12, such a local operator with window size 15 largely addresses the problem. We provide more high-resolution examples in Fig. 15.

**Efficient Implementation.** Fig. 2 in the main manuscript demonstrates the performance of LinFusion with a naive implementation. Here, we report the efficiency performance with fused operators implemented by Triton and compare the running speed with FlashAttention 2 Dao (2023) on a single RTX6000Ada GPU, as shown in Fig. 13. The conclusion is consistent with that in the main manuscript, that LinFusion achieves more significant acceleration at higher resolutions.

**Performance of Training from Scratch.** To demonstrate the potential of the LinFusion architecture introduced in this paper, we include the performance of training from scratch here. Specifically, we replace the self-attention layers in the SiT-B Ma et al. (2024a) model with the generalized linear attention layer in LinFusion and train both models on ImageNet1k-$256 \times 256$ Deng et al. (2009) from scratch for 400k iterations following the convention. Results in Tab. 11 indicate at least comparable performance of LinFusion with the vanilla attention mechanism.

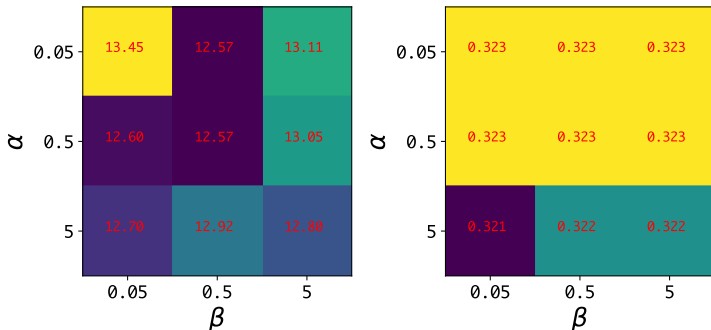

Figure 14: Results of grid search on the loss weights of knowledge distillation and attention feature matching.

| Setting | Concept-art | Paintings | Photo | Anime | Average |
|---|---|---|---|---|---|
| SD-v1.5 | 24.00 | 23.89 | **25.00** | 24.82 | 24.43 |
| w. LinFusion | **24.38** | **24.36** | 24.92 | **25.16** | **24.71** |

Table 10: Performance on the HPSv2 benchmark Wu et al. (2023).

| Setting | IS(↑) | FID(↓) | sFID(↓) | Precision(↑) | Recall(↑) |
|---|---|---|---|---|---|
| SiT-B | 146.56 | 8.15 | **5.65** | 0.72 | 0.58 |
| w. LinFusion | **157.16** | **7.09** | 5.75 | **0.73** | **0.59** |

Table 11: Performance of training from scratch. We replace self-attention layers in SiT-B Ma et al. (2024a) with the proposed LinFusion layers and train from scratch on ImageNet1k-256 × 256 Deng et al. (2009). The scale of classifier-free guidance is 1.8 here.

**Broader Evaluation.** We additionally evaluate the proposed LinFusion approach on the HPSv2 benchmark Wu et al. (2023), which measures the capability of text-to-image models given 4 various styles of generation contents. Results in Tab. 10 demonstrate the performance of LinFusion is comparable to or even better than the original SD-v1.5 model.

**Analysis of Hyper-parameters.** The distillation objective for LinFusion defined in Eq. 6 introduces two hyper-parameters: $\alpha$ and $\beta$, denoting the loss weights of knowledge distillation and attention feature matching respectively. Here we study their impacts on the final performance through a grid search. As shown in Fig. 14, we try 3 various values, 0.05, 0.5, and 5, for each of them and report the FID and CLIP-T metrics. Overall, the performance is not sensitive to the specific values of these hyper-parameters in a large range. Too small values may result in insufficient effects of these loss terms, while too large values would not benefit performance, either. The default setting, *i.e.*, $\alpha = \beta = 0.5$, is a suitable choice.

## D  LIMITATIONS

The motivation of LinFusion is to explore a linear-complexity diffusion architecture by experimentally replacing all the self-attention layers with the proposed generalized linear attention. This may not be the optimal configuration in practice. For example, it could be promising to explore hybrid structures and apply attention to deep features with a relatively smaller number of tokens but a large number of feature channels, which could be a meaningful future direction.

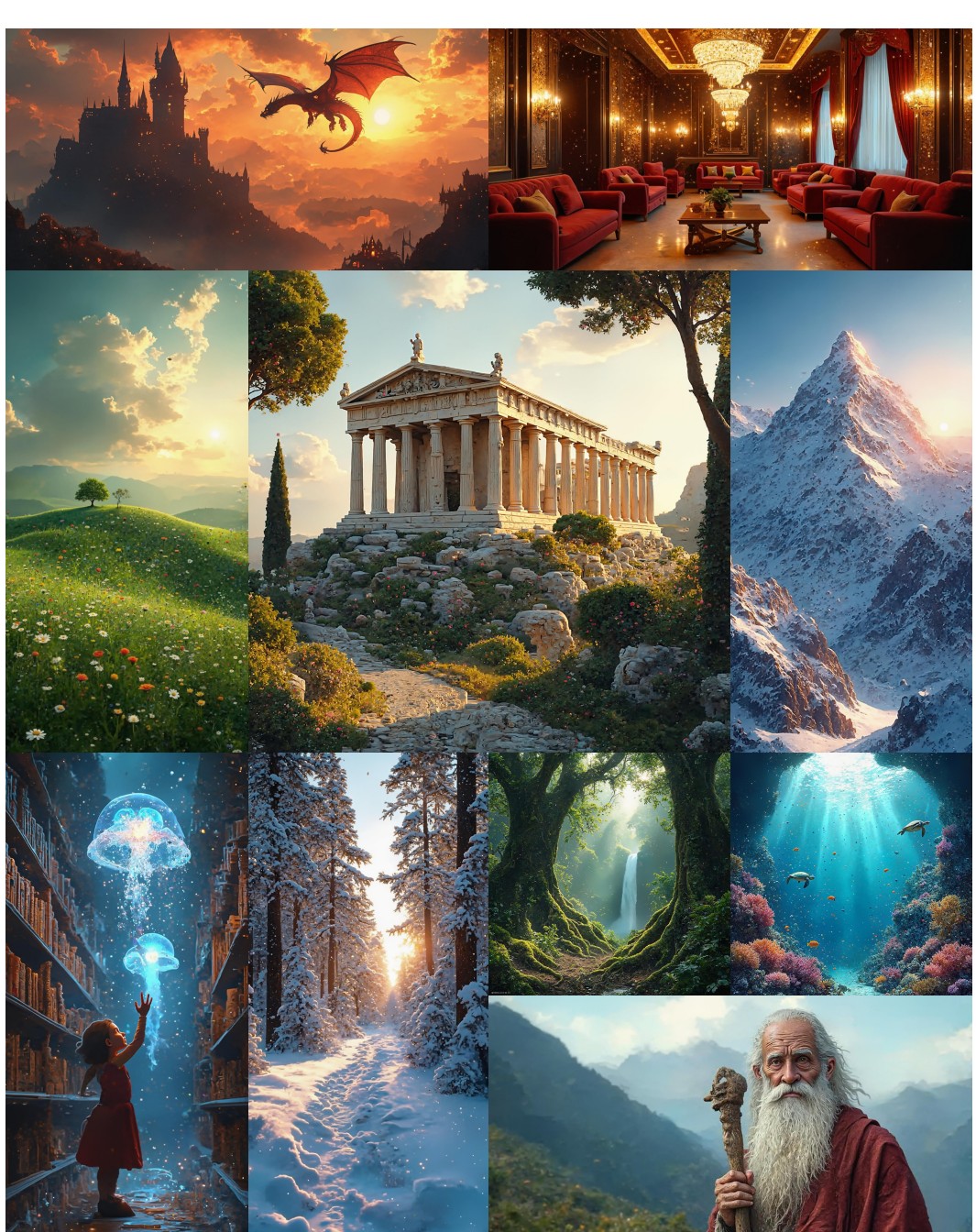

Figure 15: More high-resolution samples generated by LinFusion built on top of FLUX-1.dev.

