# OpenReview forum: "LinFusion: 1 GPU, 1 Minute, 16K Image"
_ICLR.cc/2025/Conference — Submitted to ICLR 2025_

### Official Review · Reviewer_uBFm · 2024-10-24

**Soundness:** 3
**Presentation:** 3
**Contribution:** 4
**Rating:** 8
**Confidence:** 4

**Summary:**

This paper presents a new method for efficient image generation in linear computation complex. To achieve this goal, a generalized linear attention paradigm is introduced. For training, they distill the knowledge from the pre-trained SD. This distilled LinFusion model achieves better or on-par performance than the original teacher. This model also enables many down-streaming plugins, which makes this paper far more interesting.

**Strengths:**

1. The usage of linear attention and network distillation is an interesting direction in efficient text-to-image generation.
2. Different Mamba architecture is considered and ablated, providing a reference for other related topics.
3. This paper is well-written. The detailed experiments show the advantages of the proposed method and even some down-streaming plug-and-play applications.

**Weaknesses:**

My main concern about his paper is the experiments.

1. This paper introduces three loss functions, i.e., $\mathcal{L}_{simple}$.
  $\mathcal{L}{kd}$, $\mathcal{L}{feat}$ and two hyper-parameters ($\alpha$, $\beta$). However, there is no ablation study for each of them.
2. Several down-streaming extensions (LoRA, ControlNet, etc.) have been evaluated. However, there is no further discussion of why the original network extensions work and why some of the results are better than the baseline and others are not.

**Questions:**

1. What about the influences of each loss in the training objective?
2. In L512 - L516, LinFusion uses SDEdit tricks for higher-resolution generation, and comparing with DemoFusion. Is this comparison fair? LinFusion is a specific base model. A better comparison should be to use the same settings as DemoFusion.

---

> ### Author Response · Authors · 2024-11-27
> **Response to Reviewer uBFm**
>
> We appreciate the valuable comments by Reviewer uBFm and are more than happy that the reviewer finds our idea intersting, paper well-written, and experiments detailed. We would like to address the questions raised by the reviewer as below.
>
> * > W1: This paper introduces three loss functions, i.e., $L_{simple}$, $L_{kd}$, $L_{feat}$, and two hyper-parameters ($\alpha$, $\beta$). However, there is no ablation study for each of them. | Q1: What about the influences of each loss in the training objective?
>
>   Thanks for the pertinent comment. In fact, the additional loss terms, $L_{kd}$ and $L_{feat}$, along with their corresponding loss weights are adapted from related work [a]. Thus, we do not include detailed ablation studies on the specific values of them. Nevertheless, we are willing to study their effects in this period through a grid search to the two hyper-parameters across various values. Given that computational recources required by this grid search are extensive, the experiments are still ongoing. We will upload a revision as soon as poosible on this.
>
> * > W2: Several down-streaming extensions (LoRA, ControlNet, etc.) have been evaluated. However, there is no further discussion of why the original network extensions work and why some of the results are better than the baseline and others are not.
>
>   Thanks for the insightful comment. Overall, the generalization performance of LinFusion is comparable to the original down-streaming extensions, except LCM-lora, which is not straightforward since they adopt various training objectives. That's why its generalization performance is relatively worse in this case comparing to other settings. We will include this analysis to our revision.
>
> * > Q2: In L512 - L516, LinFusion uses SDEdit tricks for higher-resolution generation, and comparing with DemoFusion. Is this comparison fair? LinFusion is a specific base model. A better comparison should be to use the same settings as DemoFusion.
>
>   Thanks for the valuable question. As shown in the first 3 rows of Tab. 5, we emperically find that combining DemoFusion together with SDEdit, without LinFusion here, may yield better performance comparing with the original DemoFusion. We thus choose to add LinFusion on this enhanced setting to validate the performance of LinFusion. Since both settings adopt SDEdit, the comparison is in fact fair.
>
>   Here, we also report the results on the original DemoFusion setting. As shown in the following table, the conclusion is consistent with that in the main manuscript, that LinFusion achieves at least comparable quality with the original model while accelerates high-resolution generation significantly.
>
>   |        Method         |    FID    |  CLIP-T   | Time (sec.) |
>   | :-------------------: | :-------: | :-------: | :---------: |
>   | Demofusion (wo Patch) |   65.44   | **0.340** |    57.56    |
>   |      +LinFusion       | **65.40** |   0.333   |    33.69    |
>
> We would like to thank the reviewer again for the constructive feedback. Our sincere hope is that our response could clear the concerns raised by the reviewer.

---

> ### Author Response · Authors · 2024-11-28
> **Revised Manuscript**
>
> Dear Reviewer uBFm,
>
> We would like to sincerely thank the reviewer for taking the time to provide valuable feedback on our manuscript. In response, we have submitted a revised version that addresses the reviewer's comments. The updates include:
>
> - Grid search on hyper-parameters.
> - Analysis of degradation on the FID performance.
> - Detailed clarifications and revisions to important sections.
>
> We are fully committed to resolving any additional issues and look forward to your continued input.

---

> > ### Comment · Reviewer_uBFm · 2024-12-01
> > **Thanks for the feedback**
> >
> > The authors' rebuttal addresses most of my concerns. After reading the revised version and the opinions from other reviewers, I keep my score unchanged.

---

> > > ### Author Response · Authors · 2024-12-02
> > > **Thanks again for the valuable review**
> > >
> > > Dear Reviewer uBFm,
> > >
> > > We are encouraged to see that most of the concerns have been addressed. We would like to express of gratitude again for the insightful comments and suggestions offered by the reviewer.
> > >
> > > Best Regards,
> > >
> > > Authors of Submission 1552

---

### Official Review · Reviewer_cB7B · 2024-10-26

**Soundness:** 3
**Presentation:** 4
**Contribution:** 3
**Rating:** 6
**Confidence:** 4

**Summary:**

This paper introduces a novel text-to-image model named LinFusion, addressing the challenge of generating high-resolution visual content with diffusion models. To optimize this, the authors propose to ultilize the popular linear attention and present methods for normalization-aware and non-causal operations, achieving performance on par with or even superior
to the original diffusion model while significantly reducing time and memory complexity. Experiments demonstrate that LinFusion achieves comparable or superior performance to the original Stable Diffusion on tasks like zero-shot cross-resolution generation, with excellent results on MS COCO.

**Strengths:**

- The paper presents an efficient text-to-image model, LinFusion, which innovatively addresses the computational inefficiencies inherent in high-resolution image generation with diffusion models.
- Two notable innovations of LinFusion are normalization-aware mamba and non-causal mamba, which significantly improving the model's performance.
- The authors have conducted an extensive set of experiments, demonstrating LinFusion's effectiveness across various resolutions and showcasing its superior capability in generating ultra-high-resolution images like 16K on a single GPU.
- The writing is clear and methodical, effectively guiding readers through the complex technical details while maintaining a focus on the practical implications of the research.
- The paper stands out for its thorough experimental validation, which not only benchmarks LinFusion against existing models but also integrates it with various components and pipelines, highlighting its versatility and compatibility in real-world applications.

**Weaknesses:**

- While LinFusion demonstrates significant improvements in computational efficiency, mamba2 is designed for language models. Could you give more comparison with state-of-the-art linear attention methods[1,2,3] in computer vision.

- The results of the experiment are unconvincing. Could provide a more holistic assessment of LinFusion's performance across different aspects of image generation., such as HPSv2，T2I_Combench，DPG？

- Could the linear attention combined with the MM-DiT blocks，which are popular in state-of-the-art diffusion models，SD3 and FLUX?

- The training cost of this method is low. Is the training sufficient? Will longer training or increasing the training resolution (1k or 2k) further improve the model effect?

[1] Cai, Han, et al. "Efficientvit: Multi-scale linear attention for high-resolution dense prediction." arXiv preprint arXiv:2205.14756 (2022).

[2] Zhu, Lianghui, et al. "DiG: Scalable and Efficient Diffusion Models with Gated Linear Attention." arXiv preprint arXiv:2405.18428 (2024).

[3] Zhu, Lianghui, et al. "Vision mamba: Efficient visual representation learning with bidirectional state space model." arXiv preprint arXiv:2401.09417 (2024).

**Questions:**

Please referring to the Weaknesses above.

---

> ### Author Response · Authors · 2024-11-27
> **Response to Reviewer cB7B**
>
> We sincerely thank Reviewer cB7B for the thorough reviews and are happy that the reviewer finds the model innovative, the method effective, the validation thorough, and the writing clear and methodical. We would like to address the concerns as below.
>
> * > While LinFusion demonstrates significant improvements in computational efficiency, mamba2 is designed for language models. Could you give more comparison with state-of-the-art linear attention methods[1,2,3] in computer vision.
>
>   Thanks for pointing some related works out. Here, we replace the self-attention layers in SD-v1.5 with them separately and get the following results on the COCO-30K benchmark:
>
>   |      Method      |    FID    |   Ours    |
>   | :--------------: | :-------: | :-------: |
>   | EfficientViT [1] |   17.54   | **0.310** |
>   |     DiG [2]      |   17.51   |   0.309   |
>   | Vision Mamba [3] |   18.36   |   0.307   |
>   |    LinFusion     | **17.07** |   0.309   |
>
>   [1] mentioned by the reviewer propose a multi-scale linear attention strategy, and we achieve comparable results with it. Note that the technical designs between [1] and LinFusion are orthogonal, it is promising to combine the methods together and use the multi-scale approach to enhance the native LinFusion module.
>
>   [2] proposes to use various scanning directions in each layer. We find that it achieves inferior results in this exerpiment since the reception field of each token is limited by the scanning order.
>
>   [3] adopts bi-directional scanning strategy like the Bi-Mamba2 studied in the manuscript. Similar to the analysis in Sec. 3.2, due to the non-causal nature of self-attention in pre-trained diffusion models, LinFusion outperforms Vision Mamba in this case.
>
>   We will include these discussion to the revision for a more comprehensive comparison.
>
> * > The results of the experiment are unconvincing. Could provide a more holistic assessment of LinFusion's performance across different aspects of image generation., such as HPSv2，T2I_Combench，DPG？
>
>   Thanks for the valuable suggestion. As mentioned by the reviewer, we evaluate the performance using HPSv2 with the images generated on the COCO-30K benchmark. The results are as follows. Overall, we achieve at least comparable and even slightly better performance with the original SD-v1.5 model.
>
>   |  Method   | Concept-art | Paintings |   Photo   |   Anime   |  Average  |
>   | :-------: | :---------: | :-------: | :-------: | :-------: | :-------: |
>   |  SD-v1.5  |    24.00    |   23.89   | **25.00** |   24.82   |   24.43   |
>   | LinFusion |  **24.38**  | **24.36** |   24.92   | **25.16** | **24.71** |
>
> * > Could the linear attention combined with the MM-DiT blocks，which are popular in state-of-the-art diffusion models，SD3 and FLUX?
>
>   Thanks for the good question. According to our experiments, directly replacing the multi-modal attention with the proposed linear attention fails to generate reasonable results. This is because the attention maps inherent in pre-trained MM-DiTs are not low-rank.
>
>   However, we reveal that, using linear attention together with sliding window attention such as [a] is a feasible solution, which also results in a linear-complexity diffusion model since the size of sliding window would not change with the increasing of image resolutions. The insight is that a large amount of attention scores are concentrated on local areas in pre-trained MM-DiTs. Local attention can handle these local interactions effectively, while global interactions are processed by linear attention. We will provide illustrative examples in the revision, which will be uploaded as soon as possible.
>
> * > The training cost of this method is low. Is the training sufficient? Will longer training or increasing the training resolution (1k or 2k) further improve the model effect?
>
>   Thanks for the insightful question. In fact, our initial exploration is conducted on SD-v1.5 and find that 50K iterations are sufficient for the training on it. We then adopt the same training configuration to other models like SD-v2.1, SD-XL, etc. Indeed, for these larger models, the training may not be sufficient. Moreover, as discussed in Lines 966~760, the low-resolution training data are naively resized to 1k scale for training SD-XL. Even in this setting, LinFusion demonstrates competitive performance. We agree with the reviewer that longer training or including more high-quality data could further enhance the performance for these more advanced models.
>
> Our sincere hope is that our response will alleviate the reviewer's concern. We are looking forward to having further discussion with the reviewer if there are remaining concerns.
>
> ***
>
> [a] Neighborhood attention transformer, Hassani et al., CVPR 2023.

---

> > ### Comment · Reviewer_cB7B · 2024-11-28
> >
> > Thanks for the detailed response and additional experiments, which have addressed my concerns. I decide to keep my score.

---

> ### Author Response · Authors · 2024-11-28
> **Revised Manuscript and Thanks for the feedback**
>
> Dear Reviewer cB7B,
>
> Thanks for the reviewer's feedback on our response, and we are glad to see that our response has addressed the reviewer's concerns.
>
> Currently, we have uploaded our revised manuscript, where points in our above response have been included:
>
> * Comparisons with more related linear attention methods in computer vision.
> * Comparisons on the HPSv2 benchmark.
> * Insights and experiments on MM-DiT-based models.
>
> Thanks again for the time and effort in reviewing our work again. We are open to continuing the conversation with the reviewer to address any lingering questions.

---

### Official Review · Reviewer_WZLt · 2024-11-03

**Soundness:** 3
**Presentation:** 3
**Contribution:** 2
**Rating:** 5
**Confidence:** 4

**Summary:**

This work introduces a generalized linear attention paradigm and extracts knowledge from Stable Diffusion to develop a distilled model called LinFusion. LinFusion avoids the quadratic increase in complexity associated with traditional attention mechanisms as the number of tokens grows, enabling the efficient generation of high-resolution visual content. Extensive experiments demonstrate that LinFusion achieves satisfactory and efficient zero-shot cross-resolution generation.

**Strengths:**

	Compared to the original SD-v1.5, LinFusion offers significant advantages in speed and GPU memory usage for generating high-resolution images.
	The extensive amount of open-sourcing and experiment reproducibility is greatly appreciated.

**Weaknesses:**

	The comparison experiments in the paper are not comprehensive; for instance, the experimental section lacks an analysis of parameters and data size.
	It is unclear whether LinFusion can outperform the latest lightweight diffusion methods, such as BK-SDM[1], on the COCO 256×256 30K dataset.
	In Table 7, LinFusion shows a significant decrease in FID scores. In contrast, LinFusion exhibits better compatibility with other components and pipelines of SD, which would be better to analyze why this occurs.
	The visual quality of the generated images does not seem particularly impressive.
[1]Kim B K, Song H K, Castells T, et al. Bk-sdm: A lightweight, fast, and cheap version of stable diffusion[J]. arXiv preprint arXiv:2305.15798, 2023.

**Questions:**

	Please address questions in "Weaknesses".

---

> ### Author Response · Authors · 2024-11-27
> **Response to Reviewer WZLt**
>
> We deeply thank Reviewer WZLt for the pertinent comments and are more than glad that the reviewer finds the improvement on efficiency of generating high-resolution images significant and the reproducibility sound. We address the concerns of the reviewer as below.
>
> * > The comparison experiments in the paper are not comprehensive; for instance, the experimental section lacks an analysis of parameters and data size.
>
>   Thanks for the pertinent comments. In fact, the parameters introduces by LinFusion are on the two MLPs handling non-linear query-key transformations, whose numbers of parameters are small. For example, on SD-XL, there are merely 25.8M parameters totally. Comparing with the original 2.6B parameters, they only contribute to less than 1% of the total parameters.
>
>   As for the data size, as mentioned in Line 396, we merely use 169K images for training, while the training data sizes for SD-v1.5 and SD-XL are 4800M and unpublished respectively. Overall, the training is much more efficient than the original models.
>
> * > It is unclear whether LinFusion can outperform the latest lightweight diffusion methods, such as BK-SDM[1], on the COCO 256×256 30K dataset.
>
>   Thanks for the comments. In the manuscript, we indeed evaluate BK-SDM and the proposed LinFusion on the COCO 256×256 30K dataset following the protocol in GigaGAN [a]. The metrics shown in Tab. 2 indicate a better performance of our approach.
>
>   Here, we additionally evaluate the performance using the official codebase of BK-SDM. The results are shown in the following table and the conclusion is consistent.
>
>   |    Method    |    IS     |    FID    |   CLIP-T   |
>   | :----------: | :-------: | :-------: | :--------: |
>   | BK-SDM-Base  |   33.79   |   15.76   |   0.2878   |
>   | BK-SDM-Small |   31.68   |   16.98   |   0.2677   |
>   | BK-SDM-Tiny  |   30.09   |   17.12   |   0.2653   |
>   |  LinFusion   | **38.20** | **12.88** | **0.3007** |
>
> * > In Table 7, LinFusion shows a significant decrease in FID scores. In contrast, LinFusion exhibits better compatibility with other components and pipelines of SD, which would be better to analyze why this occurs.
>
>   Thanks for raising the concern. In fact, the zero-shot generalization from the LinFusion trained on the original SD-v1.5 to LCM-lora is not straightforward since they adopt various training objectives. That's why its generalization performance is relatively worse in this case comparing to other settings. We will include this analysis to our revision.
>
> * > The visual quality of the generated images does not seem particularly impressive.
>
>   Thanks for pointing this out. In fact, the primary argument in the article is on the comparable performance with original models. We thus do not focus on generating impressive qualitative results.
>
>   Nevertheless, we agree with the reviewer that including more fancy results is helpful to demonstrate the capability of the proposed method. We will add a figure of image gallery in the revision for this.
>
> We would like to express our gratitude again to Reviewer WZLt for the constructive feedback to improve this article. We will definitely highlight these points in the revision, which will be uploaded as soon as possible. Please feel free to let us know if there are any follow-up suggestions.
>
> ***
>
> [a] Scaling up GANs for Text-to-Image Synthesis, Kang et al, CVPR 2023.

---

> ### Author Response · Authors · 2024-11-28
> **Revised Manuscript**
>
> Dear Reviewer WZLt,
>
> We sincerely appreciate the reviewer's valuable feedback on our manuscript. A revised version has been uploaded, incorporating the points discussed above. Specifically, we have updated the following aspects:
>
> - More qualitative high-resolution examples generated using the proposed LinFusion.
> - Discussions on the data size and model parameters.
> - Analysis of degradation on the FID performance.
>
> We remain fully committed to addressing any further concerns throughout the discussion period and look forward to your continued feedback.

---

> ### Author Response · Authors · 2024-12-02
> **Thanks again for the insightful feedback**
>
> Dear Reviewer WZLt,
>
> We would like to express our sincere appreciation for Reviewer WZLt's insightful feedback and thorough review of our submission. The comments have been instrumental in enhancing our work and refining the proposed LinFusion. Here, we would like to inquire if our response and revision address the reviewer's concerns. If not, we are fully committed to addressing any remaining issues.
>
> We deeply appreciate the reviewer's dedication throughout this process and eagerly anticipate your further feedback.
>
> Best Regards,
>
> Authors of Submission 1552

---

> > ### Author Response · Authors · 2024-12-03
> >
> > Dear Reviewer WZLt,
> >
> > We would like to extend our gratitude to Reviewer WZLt for taking the time to review our work and provide your valuable comments. As the discussion period is approaching its end, we would like to inquire if our previous response has addressed the concerns. We truly value the comments and would greatly appreciate the opportunity to discuss any possible remaining questions or clarifications. We look forward to the reviewer's response and hope to engage in further discussion.
> >
> > Thanks again for the insightful review and consideration.
> >
> > Best Regards,
> >
> > Authors of Submission 1552

---

### Official Review · Reviewer_s1Gn · 2024-11-04

**Soundness:** 3
**Presentation:** 3
**Contribution:** 3
**Rating:** 6
**Confidence:** 5

**Summary:**

This paper introduces LinFusion, a versatile pipeline designed to enhance GPU memory efficiency and boost sampling speed across various diffusion models for image generation. Specifically, LinFusion investigates recent linear-attention mechanisms to identify key factors that enable their effectiveness in diffusion models and subsequently proposes an improved, generalized linear attention to replace standard self-attention. To simplify training, the models are not trained from scratch; instead, LinFusion selectively distills its linear attention module from the original diffusion models, keeping all other weights fixed. Additional supervision is applied to align both the final output and intermediate feature representations. Extensive experiments demonstrate that LinFusion can be effectively integrated with different diffusion models, significantly accelerating image generation.

**Strengths:**

- The motivation is clear and well-founded, with a thorough analysis of existing linear attention mechanisms to identify the key factors contributing to their effectiveness in diffusion.
- Extensive experiments across various applications support the claims that LinFusion is both efficient and generalizable to different diffusion models as well as existing training and testing pipelines.
- Overall, the writing is fluent and easy to follow, with informative figures that provide ample supporting information.

**Weaknesses:**

- The comparisons are conducted only during the sampling stage. Since the proposed LinFusion module may also provide similar benefits during training, are there any metrics available for this stage?
- Related to the previous point, the paper includes only fine-tuning experiments. It would be valuable to investigate whether training a diffusion model from scratch with LinFusion replacing self-attention results in any performance drop. If so, what is the extent of this drop? Experiments on a class-conditional image generation task would be informative, even without a large-scale text-to-image model.
- The memory and efficiency comparisons primarily utilize PyTorch 1.13, which does not incorporate memory-efficient methods like flash-attention or flash-attention v2. How significant is the difference in memory consumption and sampling efficiency when these newer techniques are considered?
- The method by which LinFusion generates ultra-high-resolution images is somewhat unclear. Can LinFusion directly generate 16K-resolution images, thereby avoiding patch-wise splitting, or does it produce a lower-resolution image that is later upsampled with techniques like SDEdit?

**Questions:**

- Why does removing the patchification operation in DemoFusion in Table 5 (A -> B) increase the sampling speed? Since patchification typically reduces training and sampling costs, it seems counterintuitive.
- How the 25% of unremoved self-attention layers in PixArt-Sigma selected? Are they from shallow layers, deep layers or just randomly sampled? Can distillation twice alleviate this problem (e.g., replace 50% in the first time and train the model as described, followed by replacing the other half in the second time)?
- Minor typo: In Table 3, it should read "Bi-Directional Mamba2 w/o Gating & RMS-Norm + Normalization."

---

> ### Author Response · Authors · 2024-11-27
> **Response to Reviewer s1Gn (Part 1)**
>
> We would like to express our sincere gratitude to Reviewer s1Gn for the insightful comments and feel glad that the reviewer finds our motivation well-founded, experiments supportive, and writing fluent. We would like to address the reviewer's concerns and questions as below.
>
> * > The comparisons are conducted only during the sampling stage. Since the proposed LinFusion module may also provide similar benefits during training, are there any metrics available for this stage?
>
>   Thanks for the insightful point regarding the training efficiency. In fact, since our main method is using a self-attention-based teacher network to guide the linear-attention-based student, the training speed, in this default setting, is not distinctive. After all, the forward time for the teacher network is non-neglectable.
>
>   However, if we do not apply this distillation paradigm and follow the standard fashion of training diffusion models, there are indeed similar advantages on the training speed for LinFusion, especially when training at higher resolutions. The following table shows the backward time for a single attention layer of SD-v1.5 at various resolutions. Evaluation is conducted on a single RTX6000Ada GPU.
>
>   |    Resolution    | LinFusion | SD-v1.5 |
>   | :--------------: | :-------: | :-----: |
>   |  $512\times512$  |   0.065   |  0.042  |
>   | $1024\times1024$ |   0.074   |  0.212  |
>   | $2048\times2048$ |   0.314   |  2.028  |
>   | $4096\times4096$ |   1.192   | 28.565  |
>   | $8192\times8192$ |   4.723   | 747.047 |
>
> * > Related to the previous point, the paper includes only fine-tuning experiments. It would be valuable to investigate whether training a diffusion model from scratch with LinFusion replacing self-attention results in any performance drop. If so, what is the extent of this drop? Experiments on a class-conditional image generation task would be informative, even without a large-scale text-to-image model.
>
>   We would like to thank the reviewer for the valuable suggestion on training from scratch and are definitely willing to explore the performance in this case. Specifically, we adopt SiT [a] as a base model and replace the self-attention layers with the proposed LinFusion layers. We train and evaluate models on the widely adopted ImageNet 256x256 benchmark. As shown in the following results, the overall performances are at least compable, demonstrating the potential of the proposed linear attention method.
>
>   |  Method   |     IS     |   FID    |   sFID   | Precision |  Recall  |
>   | :-------: | :--------: | :------: | :------: | :-------: | :------: |
>   |    SiT    |   146.56   |   8.15   | **5.65** |   0.72    |   0.58   |
>   | LinFusion | **157.16** | **7.09** |   5.75   | **0.73**  | **0.59** |
>
> * > The memory and efficiency comparisons primarily utilize PyTorch 1.13, which does not incorporate memory-efficient methods like flash-attention or flash-attention v2. How significant is the difference in memory consumption and sampling efficiency when these newer techniques are considered?
>
>   Thanks for the good question regarding memory-efficient implementations. We mainly report the efficiency of naive implementations for both self-attention and LinFusion to reflect their complexities in principle. Here, we study the performance of memory-efficient implementation for a more thorough exploration. Specifically, we adopt flash-attention v2 as suggested by the reviewer for self-attention and implement the generalized linear attention in LinFusion with Triton. We find that the GPU memory consumptions are similar, since tiling strategy is adopted in flash-attention, which reduces the memory complexity to linear. For the sampling efficiency, results on SD-v1.5 at various resolutions are shown below:
>
>   |                  | LinFusion | SD-v1.5 |
>   | :--------------: | :-------: | :-----: |
>   |  $512\times512$  |   0.19    |  0.16   |
>   | $1024\times1024$ |   0.53    |  0.76   |
>   | $2048\times2048$ |   2.23    |  6.37   |
>   | $4096\times4096$ |   10.13   |  82.83  |
>   | $8192\times8192$ |   47.79   | 1902.33 |
>
>   Evaluation is conducted on a single RTX6000Ada GPU. Results indicate that the sampling speeds are comparable at relatively low resolutions. However, the acceleration by LinFusion would grow dramatically with the increasing of image resolutions.

---

> ### Author Response · Authors · 2024-11-27
> **Response to Reviewer s1Gn (Part 2)**
>
> * > The method by which LinFusion generates ultra-high-resolution images is somewhat unclear. Can LinFusion directly generate 16K-resolution images, thereby avoiding patch-wise splitting, or does it produce a lower-resolution image that is later upsampled with techniques like SDEdit?
>
>   Thanks for the question and sorry for the unclarity. In our main manuscript, we generate high-resolution images by producing lower-resolution images first and then upsampling them with SDEdit. We have demonstrated through experiments that LinFusion is also compatible with other high-resolution generation pipelines like Demofusion.
>
>   Since LinFusion is fine-tuned only on the native resolution of the diffusion model itself, it would produce repetitive patterns when generating high-resolution images directly, as shown in a lot of related works like Multifusion. Fine-tuning on larger-scale images with more computational resources could alleviate this issue, which is not the main purpose of this work and is left as a valuable future direction.
>
> * > Why does removing the patchification operation in DemoFusion in Table 5 (A -> B) increase the sampling speed? Since patchification typically reduces training and sampling costs, it seems counterintuitive.
>
>   Thanks for the good question. In fact, DemoFusion by default processes each image patch sequentially, which is slower than processing the image as a whole at the $2048\times2048$ resolution. We will add this clarification in the revision to avoid confusion.
>
> * > How the 25% of unremoved self-attention layers in PixArt-Sigma selected? Are they from shallow layers, deep layers or just randomly sampled? Can distillation twice alleviate this problem (e.g., replace 50% in the first time and train the model as described, followed by replacing the other half in the second time)?
>
>   We would like to thank the reviewer sincerely for the valuable question and raising a possible direction for exploration. In fact, we leave one self-attention layer every four layers in order for PixArt-Sigma. We will supplement this detail in the revision.
>
>   We have tried the strategy of distilling twice mentioned by the reviewer. Unfortunately, it still results in distorted local tectures. We delve into the reason by analyzing the attention maps and find that the attention interactions for pre-trained DiTs are not low-rank in fact. Thus, it is indeed challenging to replace all the self-attention layers, which the only mechanisms for token interactions in DiTs, with low-rank linear attention.
>
>   Currently, based on the above analysis, we have figured out a successful and elegant solution for adapting LinFusion to PixArt-Sigma. Specifically, we use the linear attention in LinFusion together with sliding window attention such as [b], which also results in a linear-complexity diffusion model since the size of sliding window would not change with the increasing of image resolutions. The insight is that a large amount of attention scores are concentrated on local regions in pre-trained Diffusion Transformers. Local attention can handle these local interactions effectively, while global interactions are processed by linear attention. We will provide illustrative examples in the revision, which will be uploaded as soon as possible.
>
> * > Minor typo: In Table 3, it should read "Bi-Directional Mamba2 w/o Gating & RMS-Norm + Normalization."
>
>   Thanks for the detailed inspection. We will definitely fix it in the revision.
>
> Thanks again for the thorough reviews. All these additional discussions would be included in our revised manuscript, which will be uploaded as soon as possible. We are definitely willing to interact with the reviewer if there are any further questions.
>
> ***
>
> [a] Exploring Flow and Diffusion-based Generative Models with Scalable Interpolant Transformers, Ma et al., arXiv 2024.
>
> [b] Neighborhood attention transformer, Hassani et al., CVPR 2023.

---

> ### Author Response · Authors · 2024-11-28
> **Revised Manuscript**
>
> Dear Reviewer s1Gn,
>
> We would like to express our sincerest gratitude for the valuable feedback on our manuscript. We have uploaded a revision including the above discussions. Specifically, we update the following contents in the revision:
>
> * Speed comparisons during both forward and backward time using efficient implementations.
> * Results of training from scratch.
> * Clarification and revision of necessary details.
>
> We are fully committed to addressing any remaining concerns in the rest of the discussion period. Look forward to your following feedback.

---

> > ### Comment · Reviewer_s1Gn · 2024-11-29
> >
> > Thanks for the detailed response and the updates on the revised version, which have addressed my concerns. I have no more questions at this stage and will increase my score to 6.

---

> > > ### Author Response · Authors · 2024-12-02
> > > **Thanks again for the thorough review**
> > >
> > > Dear Reviewer s1Gn,
> > >
> > > We are encouraged to see that the concerns of the reviewer have been addressed. Thanks again for the instrumental feedback to improve our manuscript.
> > >
> > > Best Regards,
> > >
> > > Authors of Submission 1552

---

### Meta-Review · Area_Chair_YwAd · 2024-12-17

**Metareview:**

This paper introduces LinFusion, a versatile pipeline designed to enhance GPU memory efficiency and accelerate sampling speeds in various diffusion models for image generation. LinFusion investigates recent linear-attention mechanisms to identify key factors contributing to their effectiveness in diffusion models. It then proposes a generalized and improved linear attention module as a replacement for standard self-attention. To simplify training, LinFusion avoids training models from scratch. Instead, it selectively distills the linear attention module from existing diffusion models, keeping all other weights fixed. Additional supervision is applied to align both the final outputs and intermediate feature representations.


This work received mixed reviews from four reviewers before rebuttal : one reviewer supported acceptance, one reviewer leaned toward borderline acceptance, and two reviewers gave a borderline rejection. After rebuttal, one reviewer raise their score to borderline acceptance and one reviewer maintain their score as borderline rejection.

After carefully evaluating the reviews, rebuttal, and refined draft, the Area Chair (AC) agrees with the concerns raised by Reviewer WZLt and the weaknesses highlighted by Reviewer cB7B, particularly regarding unconvincing experimental results. Furthermore, the quality of high-resolution generated images remains subpar, weakening the submission despite the demonstrated improvements in inference speed through linear attention.

Thus, AC recommends rejection to this work.

**Additional Comments On Reviewer Discussion:**

No

---

### Decision · Program_Chairs · 2025-01-22

Reject